# Low-Cost Sensors for Monitoring Coastal Climate Hazards: A Systematic Review and Meta-Analysis

**DOI:** 10.3390/s23031717

**Published:** 2023-02-03

**Authors:** Tasneem Ahmed, Leo Creedon, Salem S. Gharbia

**Affiliations:** 1Department of Environmental Science, Atlantic Technological University, F91YW50 Sligo, Ireland; 2Centre for Mathematical Modelling and Intelligent Systems for Health and Environment (MISHE), Atlantic Technological University, F91YW50 Sligo, Ireland

**Keywords:** coastal hazards, low-cost sensors, climate change

## Abstract

Unequivocal change in the climate system has put coastal regions around the world at increasing risk from climate-related hazards. Monitoring the coast is often difficult and expensive, resulting in sparse monitoring equipment lacking in sufficient temporal and spatial coverage. Thus, low-cost methods to monitor the coast at finer temporal and spatial resolution are imperative for climate resilience along the world’s coasts. Exploiting such low-cost methods for the development of early warning support could be invaluable to coastal settlements. This paper aims to provide the most up-to-date low-cost techniques developed and used in the last decade for monitoring coastal hazards and their forcing agents via systematic review of the peer-reviewed literature in three scientific databases: Scopus, Web of Science and ScienceDirect. A total of 60 papers retrieved from these databases through the preferred reporting items for systematic reviews and meta-analyses (PRISMA) protocol were analysed in detail to yield different categories of low-cost sensors. These sensors span the entire domain for monitoring coastal hazards, as they focus on monitoring coastal zone characteristics (e.g., topography), forcing agents (e.g., water levels), and the hazards themselves (e.g., coastal flooding). It was found from the meta-analysis of the retrieved papers that terrestrial photogrammetry, followed by aerial photogrammetry, was the most widely used technique for monitoring different coastal hazards, mainly coastal erosion and shoreline change. Different monitoring techniques are available to monitor the same hazard/forcing agent, for instance, unmanned aerial vehicles (UAVs), time-lapse cameras, and wireless sensor networks (WSNs) for monitoring coastal morphological changes such as beach erosion, creating opportunities to not only select but also combine different techniques to meet specific monitoring objectives. The sensors considered in this paper are useful for monitoring the most pressing challenges in coastal zones due to the changing climate. Such a review could be extended to encompass more sensors and variables in the future due to the systematic approach of this review. This study is the first to systematically review a wide range of low-cost sensors available for the monitoring of coastal zones in the context of changing climate and is expected to benefit coastal researchers and managers to choose suitable low-cost sensors to meet their desired objectives for the regular monitoring of the coast to increase climate resilience.

## 1. Introduction

The recent IPCC sixth assessment report has unequivocally established the human influence on the changing climate at a rate unprecedented in the last 2000 years [1]. The global energy inventory increased by 152 ZJ in 2006–2018 with respect to the 1850–1900 baseline, with 90% of this excess heat being accounted for by the ocean [2]. Thus, the global mean sea level has risen due to the thermal expansion of seawater (50%), external addition of mass to the ocean due to the melting of land-based ice (42%), and a small contribution from changes in land water storage (8%) between 1971 and 2018 [1]. Fingerprints of climate change are found in the recent weather and climatic extremes around the globe [2]. Changes caused to the oceans from past and future GHG emissions are locked in for centuries to millennia, implying that sea levels will continue to rise irrespective of the emission scenario. Therefore, important points of departure for this systematic review paper are: to recognise the various forcing factors leading to the hazards at the coast and the importance of sensors for the regular monitoring of these hazards to provide necessary data for the real-time monitoring of these forcing factors; calibrate/validate numerical models, such as storm surge models, for the development of early warning support systems; and test out the efficacy of various climate adaptation interventions such as nature-based solutions (NBS) for the attenuation of wave energy during a storm surge event [3].

Coastal zones around the world are heavily populated and developed [4,5,6,7], increasing their exposure to climate-induced hazards [4,7]. Risk is defined as [8]:Risk=Hazard ×Exposure×Vulnerability

Here, hazard is the threatening natural event (such as a storm surge) and its probability of occurrence, exposure is the asset present in the given location, and vulnerability is the damage that could be inflicted onto the exposed assets or lack of resistance to the hazard [8,9]. While the hazard is the physical aspect of the risk, the latter two come under the socio-economic aspect. This paper focuses only on the hazard.

Ref. [10], which is a key paper for this review, states that marine-hazard-related risk can be reduced through the upscaling of sustained coastal zone monitoring programs, forecasting, and early warning systems and emphasises not only monitoring the hazards (e.g., coastal flooding) and their associated drivers (e.g., sea level rise), but also the coastal zone characteristics (e.g., topography). Coastal floods, coastal erosion, and shoreline changes are some of the hazards whose risk is influenced by several forcing agents, with sea level rise being a key driver for coastal flooding [7,11], but this is not unique for coastal erosion and shoreline change and depends on several other factors [12,13]. Often, forcing factors such as high tides coincident with sea level rise and extreme weather characterised by storm surges and the associated wind-waves can drive extreme sea levels (ESLs) that will lead to unprecedented flood risk by the end of this century [7]. Climate change fingerprints are found in some of these forcing agents such as wind-waves [14,15] and storm surges [16,17], making monitoring of these factors necessary for improving the predictive skill of coastal numerical models. Routine monitoring of nearshore bathymetry/topography is equally important, even though this do not directly fall under the category of hazards and forcing factors, but instead determines the transformation of some of these forcing factors as they approach the coastal zone; for instance, offshore waves, which are monitored well by satellites, are drastically transformed in the nearshore region due to changing nearshore bathymetries and topography and contribute to increases in water levels through wave set-up and wave run-up [18]. However, these lack routine monitoring, limiting the knowledge of wave transformations at the coast and thereby limiting the ability of numerical models to accurately predict nearshore wave properties and morphodynamical changes induced by them [19].

Though the network of tide gauges has expanded around the world, it is found that only a few of those have a co-located GNSS to account for vertical land motion (VLM), which is an important factor to account for in order to measure actual sea level trends and separate the climatic signal from the non-climatic (VLM) signal [19]. Accounting for VLM is essential, as land subsidence has been shown to contribute to increased flood risks [20]. This being said, tide gauge stations are expensive to construct [21], so even though data can be retrieved at a high temporal frequency, the same cannot be said in terms of spatial coverage. Such limitations create the potential to explore low-cost solutions with performance within acceptable standards, so that such cost-effective sensors could be used to complement the existing high-cost instruments for greater spatial and temporal coverage and could also act as a backup when a standard instrument fails or becomes dysfunctional due to technical issues.

Though there are several papers focusing on satellite products for the Earth observation (EO) of coastal hazards, there are no similar papers at the time of writing this review on their low-cost in-situ counterpart, where satellite observations remain imprecise along with a low sampling frequency that could miss capturing high-frequency events [10,22]. Thus, this systematic review aims to identify within the peer-reviewed literature the low-cost sensors/sensing methods available within the last decade for monitoring coastal hazards such as coastal flooding, coastal erosion, and shoreline change, their drivers, and the physical characteristics of the coast, which are expected to be useful to the coastal researcher/manager to integrate low-cost monitoring solutions to develop climate resilience.

There is no universally agreed definition of low-cost sensors [23] and a simple Google search returns articles mainly for monitoring “air-quality” and hardly any concerned with monitoring coastal hazards. Often it is known that the cost of coastal monitoring equipment (the sensing components) is exorbitant and incurs running cost in maintenance, operation, labour charges, etc. This expense has led to sparse coastal monitoring equipment around the worlds’ coasts. Such high expenses make the establishment of a dense network of such expensive instruments to capture marine data at a fine spatio-temporal resolution cost prohibitive. Thus, this review is concerned only with the cost of the sensing component without considering the running costs incurred across the sensors’ lifespan. Given the very subjective nature of low-cost sensors, the definition adopted by [23] for low-cost air-quality sensors is loosely adapted to this review. Thus, a low-cost sensor within this systematic review can be defined as any sensor costing less than the instrumentation cost required for demonstrating compliance with national specifications for marine data (such as the tide gauges installed within the Irish national tide gauge network (INTGN) for measuring water levels and tides [24] and multi-beam echosounders fitted onto vessels such as the RV Celtic voyager [25] for bathymetric surveys).

Additionally, while [23] discusses only air quality sensors, the sensors discussed within this review are widely varying with different technicalities, which further complicates an exact definition of a low-cost sensor. For instance, low-cost sensors such as video camera systems to monitor shoreline changes and nearshore wave morphodynamics, UAVs to monitor coastal erosion and shoreline changes, and GNSS-buoys to monitor waves and sea level are different from each other not only in their technicalities but also the output data formats, even though some of these sensors may measure the same variables (e.g., shoreline change). In addition, their corresponding high-cost reference instruments are also different. Ultimately, from the coastal management/developing climate resilience perspective, the quality of data retrieved from such sensors is of great importance, which necessitates the validation of such instruments against a relevant corresponding high-cost instrument. The accuracy derived from such validation is an indication of sensor performance. Thus, in this review of low-cost sensors for monitoring coastal climate hazards, the focus is on the sensing component, the variables monitored by these low-cost sensors (coastal hazard, forcing agents, and coastal zone characteristics), and the sensor performances, without delving very deep into the technicalities (electronics and electrical theories), to give an overview of the feasibility of such sensors to monitor coastal climate hazards.

The costs of these sensors range from a couple hundred USD (DIY pressure gauge) to around USD 1000 (around USD 1400 for UAVs from the DJI phantom series commonly used in coastal erosion studies; see section on aerial photogrammetry). The sensors costing USD 1000, such as UAVs, can be considered by the adapted definition to be low-cost compared to their corresponding standard reference counterpart, for instance an airborne lidar that could cost up to several thousands of USD. Clearly, these low-cost sensors cannot be a replacement for the high-cost reference instruments, and that clearly is not the objective of this paper, but after being successfully validated against standard reference instruments, these low-cost sensors can be used to complement the sparse network of high-cost reference instruments and as a backup in case of high-cost instrument failure, and they also can be considered for deployment in ungauged locations around the world for the immediate retrieval of necessary marine data. This point can be illustrated with [26], where a GNSS-Buoy platform was assembled at around GBP 300 (GBP 100 for the buoy parts, and GBP 200 for the GNSS logger) for measuring sea levels, which is much lower than an actual tide gauge, which can costs up to thousands of USD. The validation of this low-cost GNSS buoy against a tide gauge gave an accuracy of 0.014 m (RMSE), showing that the low-cost sensor performs well. Thus, such a sensor could be explored further by interested actors for monitoring water levels in complement with extant tide gauges, as a backup when one of the tide gauges fail, and for deployment in an ungauged location for the retrieval of sea level data where the immediate set-up of a tide gauge may not be feasible. Calibration of such low-cost sensors is an extremely important topic but is not dealt with in this paper due to the lack of information regarding the same. Finally, this review is not aimed at enabling relevant actors to make decisive choices for sensor selection but rather serves as a guide and overview of the currently available low-cost sensors for the monitoring of climate-induced coastal hazards.

## 2. Methodology

The preferred reporting checklist for systematic review and meta-analyses (PRISMA 2020; [27]) was followed in conducting the present systematic literature review. The PRISMA 2020 statement consists of a 27-item checklist mainly designed for the systematic reviews of studies evaluating the effects of health interventions [27]. These 27 items are systematic steps to conduct a systematic literature review, for instance, the first five items are: to identify the report as a systematic literature review, to meet the checklist for writing the abstract, to define the rationale for the present review in the context of existing knowledge, to define objectives for the review, to specify the inclusion and exclusion criteria, and so on. The detailed checklist along with examples for each checklist is to be found at [27]. This checklist can be adapted to conduct systematic literature reviews in other fields [27], as seen from its application in tourism [28,29] and environmental studies [30]. Furthermore, [31] defined a systematic literature review protocol exclusively for environmental studies consisting of six steps that the authors named PSALSAR, which stands for protocol, search, appraisal, synthesis, analysis, and results. However, since this stems from the PRISMA protocol, the present systematic review adheres to the PRISMA protocol. Some items of the checklist such as item 11 mainly focus on assessing the risk of bias in included studies, which would have been more appropriate in intervention studies, and was not applicable within the context of this present review.

For this systematic literature review, three scientific databases, Science Direct, Web of Science, and Scopus, were searched for the relevant papers using a different combination of keywords as shown in Table 1. Only papers in English starting from 2010 to 10 November 2021 (the date of the last search) were considered. A total of 3378 papers were retrieved, and duplicates were removed to retain a total of 1804 relevant papers.

These 1804 articles were screened by title, abstract, and conclusion for relevance via a set of eligibility criteria as shown in Table 2, giving an output of 141 articles.

From Table 1 and Table 2, it can be seen that the search string and the eligibility criteria did not include all existent coastal climate hazards and forcing agents. For coastal zone characteristics, only intertidal topography was considered in this review (bathymetry below low tide was excluded due to its complexity and would need a separate review). Precipitation was excluded because the changing climate affects coastal flood hazards due to changes in the sea level more than rainfall [11]. These 141 articles were subjected to full-text screening and this was completed by 24 December 2021. Full-text articles not retrieved by 24 December 2021 (after personal communication with the respective authors via email) were excluded. These were two conference articles concerned with the beamformer evaluation of low-power coastal HF surface wave radar and the assessment of evaluation of embryo dunes using UAVs.

Following the full text screening, 60 articles were retained to be reviewed in this paper. The PRISMA diagram is shown in Figure 1.

## 3. Categories of Low-Cost Sensors for Monitoring Climate Induced Coastal Hazards

A meta-analysis of the 60 papers helped categorise the sensors into broad categories of low-cost sensors/sensing methods as shown in Figure 2. Terrestrial photogrammetry (31.7%), followed by aerial photogrammetry (26.7%), was the most commonly used low-cost sensing method. This was followed by wireless sensor networks (WSNs) (8.3%), GPS buoys (6.7%), global navigation satellite system reflectometry (GNSS-R) (5%), water level sensors (5%), ground-based beach profilers (5%), complementary methods such as terrestrial plus aerial photogrammetry (3.3%), DIY pressure sensors/gauges (3.3%), and one instance each of a high wind speed recording system, UAV-RTK Lidar, and a cable-mounted robot for nearshore monitoring (1.7% each).

This section describes in detail the categories of low-cost sensors based on three important questions:What is the low-cost sensor/sensing method used?What is/are the variable(s) derived from the low-cost sensor/sensing method, and consequently, what is the hazard or forcing agent monitored?Have the outputs of the low-cost sensors been validated? If yes, how?

### 3.1. Terrestrial Photogrammetry

The sensors/sensing techniques under terrestrial photogrammetry found from the systematic literature review could be categorised as video monitoring systems, structure from motion terrestrial photogrammetry (SfM-TP), surf cameras (surf-cams), geotagged photos, smartphone-based coastal monitoring techniques, and time-lapse photography. The following subsections describe these various sensors and sensing techniques, the variables monitored, and the validation techniques used. These sensors are summarised in Table 3.

#### 3.1.1. Video Monitoring Systems

Difficulties in monitoring nearshore hydrodynamics and morphological changes with greater spatial and temporal resolution led to the utilisation of video signals [32]. The potential demonstrated from the processing of video images to extract various nearshore and beach parameters led to the development of an unmanned coastal video monitoring system (VMS) called the Argus Station at the Coastal Imaging Lab in Oregon State University, USA [33]. Such a VMS consists of an array of cameras connected to a host computer that serves as the system control as well as a communication link between the cameras and the central data archives [33].

The Argus Station was the basic tool utilised in the CoastView project [34] to derive parameters from video images for facilitating coastal zone management [35,36]. These derived parameters were used to define a set of coastal state indicators (CSIs). Thus, video systems were evidenced to be versatile not only for scientific research but also to facilitate coastal zone management.

Coastal state indicators are defined as “A reduced set of issue related parameters that can simply, adequately and quantitatively describe the dynamic state and evolutionary trends of a coastal system” [36]. A variable derived from a video image may not necessarily be termed as a CSI, unless it can be directly used to address a coastal management problem. Thus, only variables derived from video monitoring and other monitoring methods to monitor coastal hazards/forcing agents are the focus of the review and not CSIs.

VMSs generally capture three types of images: snapshots, timex, and variance images. For instance, the Argus VMS routinely collects single snapshots, 10-minute time exposures (timex) and 10-minute variance images every hour [33]. The optical data derived from such monitoring can be used as input data or output validation for numerical models [33]. Three steps for successful monitoring with video images were mentioned by [32]: the rate of video sampling, image rectification, and the relationship between image data and parameters of interest.

Monitoring does not end at collecting digital images, but rather these images must be subjected to further processing to extract the parameters of interest, additionally requiring knowledge of camera internal and external parameters.

Following the Argus VMS, several VMSs have been developed, including Sirena [37], Cosmos [38], Horus [39], EVS [38], Kosta [40], and Beachkeeper Plus [41].

Video monitoring systems have been implemented within different funded projects, such as the STIMARE project that aims at integrating several methodologies for coastal management against erosion and flooding and monitoring coastal hazards via low-cost sensors [42]. The STIMARE project was implemented on three sandy beach sites in Italy. Ref. [43] presents the findings of a complementary approach undertaken in one of those three sites, namely Riccione in North Italy, to monitor shoreline change and wave run-up to quantify coastal erosion and flooding. They compared outputs (shoreline detection) from video images taken via a video monitoring system against the direct measurements using GPS and found satisfactory results. The post-processing was performed through image processing software in MATLAB and included rectification of the timex images to convert the image coordinates (pixels) into external ground coordinates, followed by parameter extraction, shoreline detection, and wave run-up evaluation for intense meteorological events. Ref. [44] also highlights the importance of video monitoring for validating numerical models.

Furthermore, Ref. [45] demonstrates the effectiveness of an integrated monitoring of the hydrodynamics and morphodynamics of a beach protected by low-crested detached breakwaters located in Igea Marina, Northern Adriatic Sea, Italy utilising a VMS and a 2DH numerical model [46]. The video system allowed the retrieval of real-time information of shoreline position and evolution to quantify wave run-up or coastal flooding, and the 2DH numerical model provided information on the intensity and patterns of the nearshore waves and currents. Combining the outputs from these complementary tools into a map for a potential early warning support was mentioned by the authors. Following data acquisition, a similar process was undertaken where the images, mainly the timex, were subjected to image rectification and the parameters of interest were extracted. The computed shoreline was validated with the RTK-DGPS measured waterline, showing a good correlation.

The MoZCo project is another project employing VMS to extract continuous data for the quantification of beach morphodynamics and nearshore hydrodynamics in the Portuguese coastline, which is often subjected to high-energy conditions from the ocean [47]. The data acquisition and the parameter extraction are similar to any conventional VMS system. The principal parameter extracted is the shoreline position, and as part of the MoZCo project, automatic shoreline detection algorithms were developed, which exhibited promising results when compared to manually digitised shorelines [47].

Another VMS mentioned previously is COSMOS, which is aimed at further simplifying and complementing existing video systems [38]. COSMOS has been developed in the University of Lisbon since 2007 and aims at making VMS portable, low-cost, robust and easily installable. Ref. [38] gives a detailed explanation of this. Though the modules attached to this VMS are like any other VMS, its simplification lies in the fact that the image acquisition and the image processing tasks are detached to make the system, camera, and computer independent. This feature facilitates the use of any type of camera, including standard non-metric cameras. A standard IP surveillance camera is used in COSMOS and the acquired data are stored on an onsite hard disk, making the system portable. Following data acquisition, image rectification is preceded by camera calibration and image correction, which is explained in detail by the authors and performed using the Rectify Extreme program available on the COSMOS website [48]. Following rectification, timex and variance images are created using a tool called COSMOS IPT [48], following which a TIFF world file is automatically created to be imported into standard GIS applications such as ArcGIS [49] for the extraction of the desired parameters. This system has been successfully applied across several sites in Portugal, Poland, and Italy to study beach morphologies such as intertidal topography, coastline evolution, inlet migration, storm-induced morphological change, beach nourishment evolution, and nearshore hydrodynamics such as wave breaking and dissipation patterns. Future developments in COSMOS will focus on the design of a communication infrastructure that will enable utilisation of COSMOS in a real-time coastal hazard warning system.

The outputs of Argus VMS have also been used to generate a digital elevation model utilising principal component analysis to quantify the changes in local altimetry to identify areas of erosion and accretion [50].

In addition to employing video systems for integrated monitoring [45], the outputs from such systems can be used to complement remotely sensed data from satellite and other terrestrial photographs such as crowd-sourced photos [39]. Ref. [39] also elaborates on different tools such as C-Pro [51] and SHOREX [52] to derive shoreline position from terrestrial and satellite images, respectively.

The techniques of extracting the main product from these video images, that is, shoreline position, are evolving. Ref. [53] employed a deep learning algorithm called mask R-CNN to automatically extract the waterlines (or shorelines) from thousands of timex images captured by three VMSs in three different macrotidal beaches in the Normandy coastline for quantifying the intertidal topography. This intertidal topography was validated with a digital elevation model (DEM) for each of those three sites using a differential global navigation satellite system (DGNSS), showing that the shorelines extracted using mask R-CNN are reliable. A methodology was proposed in [54] to extract the shoreline position from VMS derived images based on sensor fusion principles for automatic shoreline detection. The automatically detected shoreline was validated with manually digitised shorelines by three expert users, revealing satisfactory accuracy.

The parameters derived from the video images to monitor the coastal hazards/forcing agents along with general information such as principal hardware specifications and the software used to extract these variables are listed in Table 3.

#### 3.1.2. Structure from Motion-Terrestrial Photogrammetry (SfM-TP)


Structure from motion (SfM) is a process of reconstructing 3D structures from 2D images taken from different viewpoints [55], in which the geometry of the scene, camera position, and orientation are automatically solved without the requirement of an a priori network of targets with known 3D positions [56].

However, since the raw SfM output is in a relative coordinate system, its transformation to the absolute coordinate system to extract metric data requires establishing a network of ground control points (GCPs) [56,57]. SfM photogrammetry has been evidenced as useful as a low-cost tool in geoscience applications to retrieve complex topography with decimetre-scale vertical accuracy [56].

Terrestrial photogrammetry has been successfully applied to quantify erosion on coastal cliffs [58], and terrestrial photogrammetry combined with SfM has been used to monitor coastal cliff morphological changes by deriving high-resolution topographic data of the site monitored [59,60,61,62].

A comparative approach to monitor a receding coastal cliff (cliff foot and cliff face) in Torre Bermeja (Spain) was carried out using aerial photography, SfM-TP, and a terrestrial laser scanner (TLS), and it was found that SfM-TP, which was carried out using a consumer-grade camera, gave similar outputs compared to the more expensive TLS. Fourteen TP surveys were conducted within a duration of eight months in two sections of the cliff; subsequently, 28 point clouds were generated in Agisoft PhotoScan software (now Agisoft Metashape) [63]. The geomorphological change was quantified by comparing the 28 point clouds generated from the surveys utilising the M3C2 algorithm [64] in an open source software called CloudCompare [65].

Furthermore, the high-frequency and low-cost surveys utilising SfM allowed the quantification of cliff retreat in response to meteorological events, that is, maximum cliff foot retreat occurred due to the superposition of a medium-scale storm with high tidal range. The only drawback in utilising SfM was the time-intensive processing of images to generate the 3D point cloud [56,59], which may take between 7 and 56 h on a 64-bit system with 2.8 GHz CPU for 400–600 images with 2272 × 1740 pixel resolution [57].

Similarly, Ref. [60] employed this SfM technique to quantify the coastal cliff geometry of an eroding cliff on the coastline of Stara Baška on the island of Krk, Croatia. Here, unlike the previous study, the morphological change of the eroding cliff was not tracked over a period, but rather the topography was used as a baseline for future assessment of coastline evolution and coastal zone management. From orthorectified aerial photographs over a few years, the coastal cliff in Stara Baška was seen to retreat by 5 m. The photogrammetric survey was conducted in a single day with a consumer-grade camera and the site was categorised into several sectors. Multiple overlapping photographs were taken from various perspectives and a point cloud was generated using Autodesk ReCap online service (not available on the website mentioned in [60], instead refer to [66]). Finally, the CloudCompare software was used to georeference and combine all the point clouds into a single large-scale model for the entire site. The output was validated for a specific sector of the site against some RTK-GPS points, showing good model performance.

Ref. [61] evaluated the potential of a fixed multi-camera array for the reconstruction of retreating coastal cliffs and investigated the effects of camera height and obliqueness on image reconstruction, making use of an action camera (GoPro Hero4 black), which is unconventional in the sense that the large field of view (FOV) incorporates greater radial distortion; however, Ref. [61] found that this could be compensated for with the use of commercial software such as Agisoft Metashape, generating point clouds that were at times even better than the TLS survey carried out for validation.

In all of the above studies, GCPs were used to transform the reconstructed 3D image in world coordinates, but [62] carried out the 3D reconstruction of a coastal cliff with direct georeferencing without using GCPs, recognising the time-consuming procedure of setting up a GCP network and measuring their positions [67,68]. The RTK-GNSS assisted SfM photogrammetric method using a reflex and a smartphone camera was deployed in the Porsmilin beach on a macrotidal coast in Brittany, France to monitor the cliff located to the west of the beach. The measurement system consisted of a custom-built wooden frame to mechanically connect the mobile antenna of the RTK-GNSS receiver and a camera, which was mounted onto a tripod for photography. Photographs were taken 50 m from the cliff face in a fan-shaped capture along different camera stations to span the entire cliff, with every station capturing approximately 5–10 photographs. The spatial resolution of the images was 1.60 cm/pixel for the reflex camera and 1.93 cm/pixel for the smartphone camera. For the smartphone camera, as it came equipped with an internal GNSS, the image geotags were also additionally tested for accuracy along with the RTK-GNSS georeferenced images. The processing was carried out in Agisoft Metashape and all the images from both the cameras were subjected to the same parameters to avoid biases. The final product was a coloured point cloud exported for further analysis. However, for the smartphone camera, two processes had to be carried out: one with the camera position file using the RTK GNSS and the other with the camera position file from the camera’s internal GNSS. A comparison between these two camera position files resulted in a RMSE of 3.44 m, which the authors state is due to the low precision of the internal GNSS module. The point clouds obtained from the RTK-GNSS assisted SfM photogrammetry were validated for accuracy against a TLS point cloud. The low precision of the cameras’ internal GNSS module translated into low accuracy when the corresponding point cloud was compared against the TLS point cloud (mean error = 0.10 m). The comparison of the point clouds from the reflex and smartphone camera (utilizing the RTK-GNSS) with the TLS point clouds found accurate measurements for both cameras (see Table 3). The larger deviations from the TLS point cloud were mainly over the vegetated areas. Additionally, the comparison of the reflex and smartphone camera point clouds revealed strong agreement (mean error = 0.5 cm). The authors state that such a monitoring technique is transposable to citizen science.

Terrestrial SfM photogrammetry can be greatly leveraged by utilising platforms like UAVs without the constraint of terrain. SfM photogrammetric approaches utilising UAVs are discussed in detail in Section 3.2.

#### 3.1.3. Surf Cameras

The potential of existing networks of surf cameras (surfcams) for shoreline monitoring [69,70] and nearshore wave measurements [69] were examined. The nine surf camera (surfcam) sites for both studies were the same, located within seven sandy embayments along 250 km of microtidal New South Wales, Australia. The surfcam network utilised in the studies consisting of pan-tilt-zoom IP cameras were managed by Coastal Conditions Observation and Monitoring Solutions (CoastalCOMS, Varsity Lakes, Australia) [71] and were mounted inside small unobtrusive buildings, usually surf club buildings. The data acquisition, storage, and analysis were carried out using Amazon Cloud (Amazon Web Services, Seattle, WA, USA). The surfcam-derived shorelines were validated against the hourly (daylight) Argus video monitoring system-derived shorelines and RTK-GPS measurements in South Narrabeen, Australia, whereas in the remaining eight sites the shorelines were only compared against monthly RTK-GPS data. The image analysis technique on the surfcam-derived images was developed by CoastalCOMS to detect shorelines. Their image rectification technique is referred to as the “transect technique”, which [69] states is limiting as it does not account for changes in the vertical plane that could translate to large horizontal errors, reducing the accuracy of the detected shorelines. Ten months of daily surfcam and Argus-derived shoreline data were compared, showing a general similarity in the variability, but the surfcam shorelines were landward-biased. The authors hence applied a simple geometric correction to correct for induced errors, following which the RMSE improved for the transects near the surfcam and not for the distant oblique transects. Similar results were seen upon comparison to the monthly RTK-GPS data, with the RMSE and $R^{2}$ improving upon the application of the geometric technique. Ref. [69] also investigated the capabilities of the surfcams to monitor nearshore wave parameters using the Wave Pack system by [72]. The significant wave height (average of the highest one-third of all waves) and the period were derived through the Wave Pack system from surfcams at two sites with co-located buoys. The weak statistical relationship with the buoy data ($R^{2}$ < 0.6) and the systemic over-estimation of smaller waves and under-estimation of larger waves attributed to possible rectification error and beach/wave type led to the conclusion that the quantitative wave monitoring abilities of the surfcams were inadequate and that shoreline monitoring was more successful following the introduction of a geometric correction that significantly improved the accuracy of the monitored shorelines.

Ref. [70] provided a more objective approach to [69] for the exclusive evaluation of the shoreline monitoring capabilities of the surfcams at the same sites. The validation techniques were also the same. Ref. [70] applied two different geometric transformation techniques to the surfcam operators’ “transect technique”, thereby demonstrating significant improvements in the statistical metrics in comparison to the Argus-derived shorelines in South Narrabeen and RTK-GNSS measurements across all the nine sites. Amongst all the sites, the sites with the highest camera angles relative to the beach profile returned more accurate results (SD of errors ~1–2 m) than the sites with a lower elevation angle (SD of error ~3–4 m). This was attributed to the fact that the visibility of the shoreline could be obstructed by morphological features in the foreground for low-elevation cameras such as surfcams, thus establishing the importance of the camera’s elevation with respect to the beach slope and width. For the sites with the highest surfcam elevation, the implementation of the geometric corrections led to levels of accuracy that suggested that the opportunistic use of surfcams can be sufficient to complement and extend the shoreline monitoring capabilities of more sophisticated coastal imaging systems.

#### 3.1.4. Geotagged Photos

Ref. [73] repurposed existing infrastructure (street signs) as erosion pins [74] to study storm-induced erosion of the beach dune system in Isle of Palms, a mixed-energy coast in South Carolina, USA. A pre-storm survey was carried out utilising a GPS camera to derive geotagged photos of the existing street signs, which were embedded along the dune system to mark the location of adjacent avenues, followed by a post-storm survey of the same geographic area. From each geotagged photo, a point shapefile of the street signs was generated and stored along with the corresponding photo in the ArcGIS geodatabase for pre- and post-storm survey. The morphological change or erosion–accretion dynamics were quantified as the relative change per pole between the pre- and post-storm photos. This relative change in distance was determined from the distances between the holes or perforations, perforation distance (PD), present in the street sign, and the number of holes visible in the photo. In the cases where the pre- and post-storm photos had an unobstructed view of the street sign, the distance (D) from the bottom of the sign to the bed can be expressed as:(1)$$D_{\mathrm{pre}}=\mathrm{HC}*\mathrm{PD}$$
where HC = hole count and PD = perforation distance. Similarly, Equation (1) can be applied to the post-storm photo to calculate D_post_, and this difference represents the relative change in centimetres.
(2)$$\Delta_{\mathrm{rel}\;}=D_{\mathrm{pre}}-D_{\mathrm{post}}$$

However, these street signs may be obstructed by vegetation or completely uprooted by the storm. In such cases, the methodology to calculate relative changes was modified and further elaborated in [73]. These relative change values were subdivided into six qualitative categories from “no change” to “extreme change” indicative of the erosional intensity. The authors identified the potential of crowd-sourced geotag photos to make this monitoring time-efficient and identify the scalability of this process to any developed coasts (with existing infrastructure) and to the study of other phenomena such as dune recovery post-storm. The authors state that this technique is robust; however, no validation study was carried out.

#### 3.1.5. A Smartphone-Based Technology for Coastal Monitoring

Ref. [75] carried out a detailed accuracy analysis of the potential of the smartphone to monitor shorelines in Gwangalli Beach, Busan, South Korea as a much lower-cost alternative to the conventional VMS (e.g., Argus VMS). The obtained results were validated against TLS DEM data. Furthermore, the results were also compared with metric cameras. The image acquisition was automated using an app that collected 6 images at 10 s intervals every 30 min, yielding timex images. The camera intrinsic and extrinsic parameters were determined using a Rollei-metric Close-range Digital Workstation (CDW) and the ERDAS Leica Photogrammetry Suite (LPS; renamed to “IMAGINE photogrammetry [76]), respectively. Using a total station, 60 GCPs were established. A DEM was generated using a TLS, which was used for the orthorectification of the smartphone-camera-derived timex images to generate an orthorectified time exposure image. Shorelines from the orthorectified photos for different time stamps were used to create an intertidal DEM, which was then compared against the TLS DEM. The experiment was repeated for different camera calibration scenarios and improvements in results were obtained with proper camera calibration, establishing the importance of proper camera calibration to correct for the lens distortion in non-measuring camera lenses. This study, along with that by [62], establishes the usefulness of smartphone cameras for coastal erosion monitoring.

#### 3.1.6. Time-Lapse Photography

Ref. [77] utilised time-lapse photos to quantify shoreline erosion and coastal water levels in two low-lying western Alaskan communities that are vulnerable to storm surge impacts due to their inability to assess the risk from such hazards due to the lack of monitoring equipment and updated baseline data. Such data gaps make monitoring the hazard and its driving factors difficult. The water levels were measured in Whittier, Alaska using a time-lapse camera for 6 days in March 2017. The FOV of the camera included the water surface, as well as a ladder with fixed distances of 0.305 m between the rungs extending into the water.

The time-lapse photos were processed in MATLAB. For converting the pixels into real-world metrics, a vertical datum was selected on the photo for the measurement of water levels. Water levels were read with respect to a local elevation datum in the photo. The known distances between the ladder rungs were used to calibrate the pixels as shown in Equation (3):(3)$$m_{pix}=\frac{y_{dist}}{abs\left(y_{1\;}-y_{2\;}\right)}$$
where *m_pix_* is the conversion from pixels to metres, $y_{dist}$ is the known user-entered distance (in metres) between two points on the photo, and $y_{1\;}$and $y_{2\;}$are the pixel values of increasing y distance from the picture origin of the selected points. Finally, the water levels were calculated as follows:(4)$$Z_{wl}=m_{pix}\left(Z_{datum}-Z_{select}\right)$$
where *Z_wl_* is the elevation of water level in metres relative to a vertical datum, *Z_datum_* is the elevation of the selected local datum in pixels, and *Z_select_* is the elevation of the selected water level in pixels. The water level values were exported to a data matrix along with the date time stamp. These measured water levels were validated with a laser-telemetered unit called the iGauge system [78]. Because the measurements were not over the same time interval, the photo-measured water levels were linearly interpolated to iGauge-measured time values for direct comparison. The RMSE was found to be 0.14 m.

A similar method was adopted to monitor shorelines in Port Heiden, Alaska, where the time-lapse camera was installed at the edge of an eroding bluff (proxy for the shoreline). The time-lapse photos were subjected to the same methodology undertaken to calculate the water levels. The photo-derived shoreline values were validated against ground measurements using measuring tape by environmental coordinators. An RMSE of 0.44 m was obtained. Additionally, the time-lapse photos also helped capture the several storms characterised by sea surface wave activity during the monitoring period, which [77] states is an added advantage, as it can help determine if the increased water levels were due to storms or other competing factors such as tidal regimes or river run-off.

Ref. [79] also made use of time-lapse photography to monitor the dynamic morphological changes in the intertidal zone of a sandy beach in a high-energy exposed coast of NW Ireland. A fixed time-lapse camera was obliquely oriented at a beach dune system for 3 months in the NW of Ireland to monitor the lateral movements of intertidal bar and dune edge dynamics. The camera captured eroded frontal dunes, the dune toe, upper beach, and an intertidal area for at least the first 150 m of the beach length. The time-lapse camera (TLC) was programmed to take photos every 30 min during daylight hours. Ref. [79] demonstrated the effectiveness of the TLC by quantitatively calculating intertidal bar migration and stated that such a technique would facilitate comparing forcing factors driving changes in intertidal dynamics similar to [77]. The technique to calculate bar migration consisted of setting up GCPs along a transect for image calibration and the leading edge of the bar was taken as the reference for calculating the migration. The authors state that such a technique could be used qualitatively for wave run-up and dune toe encroachment information during high-energy storm events. The total change in the morphology (31.23 m) was accurate when compared to the DGPS measurement of 30 m over the same period.

**Table 3 sensors-23-01717-t003:** Summary of papers on terrestrial photogrammetry that includes type of paper, sensors used, parameters derived, software used to extract parameters, hazards/forcing agents monitored, validation, and accuracy.

Authors	Journal/Conference	Principal Sensing Components Used	Variables(s) Derived	Software Used to Extract Variables(s)	Hazard/Coastal Zone Characteristic/Forcing Agent Monitored	Validation	Accuracy
[38]	Journal	IP-video MOBOTIX	Wave characteristics, shoreline position, intertidal topography	Rectify extreme, COSMOS IPT, ArcGIS	Coastline change, storm-induced morphological changes and inlet migration/intertidal topography	GCPs measured using RTK-GPS	Positional accuracy decreases with increasing distances from the camera with greater error in the alongshore direction (RMSE = 9.93 m) than in the cross-shore direction (RMSE = 1.18 m). The RMSE for the swash line was 1.4 m, and for the intertidal topography the vertical RMSE was found to be 0.08 m
[43]	Conference	Raspberry Pi and 8 MP camera	Shoreline position and wave run-up	Not mentioned	Coastal erosion and coastal flooding	Not mentioned	Not mentioned
[44]	Journal	Raspberry Pi and 8 MP camera	Shoreline position and wave run-up	Not mentioned	Coastal retreat and coastal flooding	GPS measurements of the shoreline	Bias = 0.14 m, RMSE = 1.41 m
[45]	Journal	Super HAD CCD ½’’ and two 8-megapixel digital still cameras; Olympus SP500 UZ	Shoreline position and wave run-up	Not mentioned	Shoreline change and flooding	Computed shoreline validated with RTK-DGPS measurement	Not mentioned
[47]	Conference	Video station with low-cost cameras-no specificities mentioned	Shoreline position	MATLAB/ArcGIS	Coastal erosion	Comparison with manually digitised shorelines	Average = 2.7 pixels, sd = 2.2 pixels
[50]	Conference	ARGUS VMS	Digital elevation model	MATLAB and Erdas-Image	Vertical changes (erosion/accretion)	In situ field altimetry data	Largest biases = 0.18 m–0.22 m, R squared = 0.80–0.92
[51]	Thesis	Not mentioned	Shoreline position	C-Pro/SHOREX	Coastal evolution	Not mentioned	Not mentioned
[54]	Journal	Not mentioned	Waterline	Not mentioned	Coastline change/intertidal topography	Intertidal topography validated with DGNSS measurement	RMSE = 0.22 m–0.33 m, R squared = 0.93–0.99
[55]	Journal	Video station with 5–6 cameras-no specificities mentioned	Shoreline position	Not mentioned	Coastal erosion	Comparison with manually digitised shorelines	RMSE = 1.7 m, bias = –0.03 m
[59]	Journal	Panasonic Lumix DMC-FZ 1000 camera	3D point cloud	Agisoft Metashape, CloudCompare	Coastal cliff changes	Terrestrial laser scanner	Non-significant differences in the point clouds of SfM-TP and TLS
[60]	Journal	Ricoh GR Digital IV camera	3D point cloud	Agisoft Metashape, CloudCompare	Coastal cliff changes	RTK-GPS	RMS of vertical offset = 7 cm, RMS of horizontal offset = 6 cm
[61]	Journal	GoPro Hero 4 Black action camera	3D point cloud	Agisoft Metashape, CloudCompare	Coastal cliff changes	Terrestrial laser scanner	Mean difference = 4–10 mm, sd = 5.30–9.69 mm
[62]	Journal	Nikon D800 Reflex camera/Huawei Y5 2016 Smartphone, Topcon^®^ HiPer V GNSS receiver	3D point cloud	Agisoft Metashape	Coastal cliff changes	Terrestrial laser scanner	For Nikon D800: mean error = 0.03 m, sd = 0.047 mFor Huawei Y5 2016: mean error = 0.02 m, sd = 0.038 m
[69]	Journal	Existing surfcam network	Shoreline	Propriety software used by CoastalCOMS	Shoreline change	RTK-GPS	RMSE = 4.4 m–16.4 m and $R^{2}$ R squared = 0.58 m−0.91 m at the 9 sites after geometric correction
[70]	Journal	Existing surfcam network	Shoreline	Propriety software used by CoastalCOMS	Shoreline change	RTK-GPS	Cross-shore error < 1 m and sd = 1–4 m
[73]	Journal	Olympus TG-860 GPS camera	Sand level variation from street signs (used as ad hoc erosion pins) from geotagged photos	ArcGIS	Coastal erosion	Not mentioned	Not mentioned
[75]	Journal	Smartphone; Samsung Galaxy S	Shoreline	Not mentioned	Shoreline change	Terrestrial laser scanner	Vertical accuracy; sd = 0.037 m
[77]	Journal	Time-lapse camera	Shoreline and water level	MATLAB	Shoreline change and storm surge	Water level validated against iGauge and shoreline validated against tape measure	RMSE for water level = 0.14 m and for shoreline = 0.44 m
[79]	Journal	Time-lapse camera; Brinno TLC200	Intertidal bar and dune edge used as shoreline proxy	Not mentioned	Bar migration and cliff changes	DGPS	Not mentioned

### 3.2. Aerial Photogrammetry

Aerial photogrammetry spans the methods to capture information from an aerial platform such as manned aircraft, UAVs, blimps, balloons, and kites. The low-cost types that are widely discussed in the scientific literature, mainly unmanned aerial vehicles and kite aerial platforms, are discussed in this section.

#### 3.2.1. Unmanned Aerial Vehicles

Unmanned aerial vehicles (UAVs), also known as unmanned aerial systems (UAS), remotely piloted aerial systems (RPAS), remotely piloted aircraft (RPA), aerial robots, or drones [80], are unmanned powered aircraft that can be operated automatically/semi-automatically with preprogramed flight planning [81] and have been widely deployed for geoscience applications. UAVs have emerged as a feasible remote sensing alternative to the conventional manned aerial platforms in terms of lower cost, increased operational flexibilities, and greater versatility [81] and can be classified into two distinct categories: fixed wing and rotary wing [82]. The parameters of interest are extracted from the imagery captured via sensors attached to the UAV. For a brief overview of the various aerial platforms for coastal and environmental remote sensing, refer to [81]. Refs. [82,83,84] provide comprehensive reviews on different aspects of these aerial platforms. Ref. [82] follows the development of such platforms from the earliest RC model aircraft to the present day ready-to-fly (RTF) UAVs. The authors review the different types of UAV platforms, such as fixed wing and multi-rotor, as well as the variety of sensors that can be fitted onto these platforms. They cite several journal papers from 2014 to 2016 highlighting the widespread applicability of UAVs for coastal and marine applications and, additionally, provide three examples to illustrate the potential of UAVs. Ref. [82] also enumerates the commercially available and open-source software for the processing of UAV data.

Ref. [83] reviews the UAVs from a coastal zone management perspective, reviewing papers on topics such as aquatic vegetation, coral reefs, long-term beach morphology, coastal flooding and erosion, land cover mapping, etc. The authors also provide sections summarising the UAV sensors, software, and validation techniques used. Ref. [84] based its review on materials and methods used to monitor coastline evolution in the period 2000–2019. Their review has five main categories, namely: period of interest; type of data acquired; shoreline spatial extraction methods, indicators, techniques and models; software; and erosion/accretion estimations methods and algorithms. They identified the growing potential of UAVs as a low-cost high-resolution image acquisition tool and summarised the most commonly used software for UAV data processing.

Reliance on high-resolution topographic data is paramount for quantifying erosion due to storms, tidal action, etc. As deployment of UAVs during a storm is often difficult, the reliance is on pre- and post-storm beach surveys to quantify the impact of a storm event. This necessitates the availability of a baseline beach morphology against which storm impacts may be gauged. Ref. [85] argues that beaches that are subjected to high-energy conditions require extreme storms to cause significant morphological impact. The focuses of the UAV studies from the retrieved peer-reviewed papers are coastal erosion monitoring and shoreline changes.

The UAV studies mainly consist of taking images using a red–green–blue (RGB) camera mounted onto a suitable UAV platform and processing those images using a SfM workflow within a suitable software to derive DSMs/DEMs/orthomosaics [67,86,87,88,89,90,91,92,93,94,95,96]. Although most authors used only a single type of UAV for conducting the studies [86,87,89,90,91,92,93,94,95,96], others used different types of UAVs [67] and different models of the same UAV [88] to discern any differences in the output induced due to different UAV types/models. While [67] found that the markedly lower cost quadcopter outperformed the more expensive fixed-wing UAV with DSM accuracy values comparable to a reference lidar DEM and the ground truth data, Ref. [88] found that a more recent version of the same UAV quadcopter fitted with a 20 Mpix camera resulted in a substantially more accurate DSM and that the DSM from the lower-cost version largely met the standards for coastal monitoring.

GCPs are used for georeferencing, and though it is possible that the aerial images are already geolocated by the UAV’s autopilot, the utilisation of external GCPs lead to more accurate results (lower standard error) of the DSMs/DEMs/orthomosaics [67,86,93]. Additionally, Ref. [88] also demonstrated a positive correlation between number of GCPs and DSM vertical accuracy.

Flight height is an important parameter as it determines the ground sampling distance (GSD), which is defined by [93] as:(5)$$\mathrm{GSD}=\frac{H*S}{F*N}$$
where F is the sensor focal length, H the flying altitude, S the sensor size, and N the number of pixels of the sensor. GSD is the linear distance between two consecutive pixel centres measured on the ground; a higher value of the GSD represents a lower spatial resolution of the image [97]. Thus, from Equation (5), for fixed sensor parameters, the flight height would determine the resolution of the photogrammetric outputs. Additionally, even while flying at a constant height, the captured images may not have the same GSD due to terrain elevation differences and changes in camera angle while shooting, but since the DSMs and the orthomosaics are created using the 3D point cloud and camera positions, an average GSD is computed and used [97]. The UAVs were flown at different heights such as 50 m [93], 60 m [89,92], 65 and 85 m [88], 90 m [67], 100 m [90,94,95], 120 m [87], and 150 m [86,96], yielding different resolutions of the aerial images. The authors also used different ratios for the front/side overlap of the aerial images during flight planning such as 60%/40% [86], 85%/75% [87,95], 80%/50% and 75%/55% [88], 80%/70% [89,96], 70%/70% [67,90], 60%/60% [91], 70%/60% [92], 85%/70% [93], and 75%/75% [94]. Most authors except for [93] used different software such as ArduPilot [67,86,98], MikroKopter OSD Tool [87,99], Litchi application [88,100], Pix4DCapture [89,90,95,101], MapPilot app [91,94,102], and DroneDeploy web platform [93,103] for automating the UAV flights.

Quantification of morphological changes due to forcing agents such as storm winds and high-energy waves was carried out by conducting pre- and post-storm surveys of the area and generating pre- and post-storm DSMs/DEMs by implementing the SfM algorithm within software such as Agisoft Metashape [86,88,89,91,92,94,96] and Pix4Dmapper [67,87,90,93,95,104] and taking the difference of the pre- and post-storm DSMs/DEMs to give the DSM/DEM of difference or DoD [86,87,88,89,90,93,94,96]. DoD > 0 would indicate an area of deposition and DoD < 0 would indicate an area of erosion. Therefore, when applied over the entire beach face, the total erosion volume was estimated by summing the DoD over the area where it was negative [86]. Similarly, the total deposition volume was estimated by summing the DoD over the area where it was positive. Limit of detection (LoD) was employed to remove areas that may not have experienced any change yet exhibited small differences in the elevation due to the uncertainty in the original DSMs [86]. For details on generating DSMs/DEMs/orthomosaics from UAV images using computer vision algorithms, refer to [105]. Shoreline changes could also be quantified from the orthophotos/orthomosaics generated using the SfM workflow [88,89,91] in addition to calculating volumetric changes.

The accuracy of quantifying the volumetric changes and shoreline changes depends on the accuracy of the generated DSMs/DEMs/orthomosaics. To validate the DSMs/DEMs, vertical accuracy is calculated by comparing the measured (using for instance RTK-DGPS) vertical values of a set of individual checkpoints (ICPs) with the computed vertical values obtained in the 3D models obtained at the same horizontal coordinates. The authors used different statistical metrics to express the accuracy such as the root mean squared error (RMSE), coefficient of determination ($R^{2})$, standard deviation, optimising median absolute deviation (nmad), etc., and are elucidated in Table 4. As evident from the high accuracy values in Table 4, UAVs are effective tools for coastal erosion monitoring facilitating the quantification of volumetric sediment losses from beaches and shoreline changes, thereby providing valuable insights into erosional trends for a given area; for instance, [90] observed that open-ocean beaches mobilise three times as much sediment as embayed beaches in the high-energy SE Australian coastline and distinguished between slowed and accelerated erosional modes.

The effectiveness of the UAVs for coastal monitoring is further reinforced from three comparative studies where the outputs of UAV photogrammetry are compared to high-cost references such as total-station, RTK-GNSS, TLS, and terrestrial and airborne lidar [93,94,95], exhibiting high accuracy values as seen from Table 4.

#### 3.2.2. Kite Aerial Photography

The usage of drones can be hindered by strong winds and dust, especially in sandy coastal environments [106]. An alternative to that could be kite aerial photography [107].

Ref. [108] utilised kite aerial photography with an SfM-MVS workflow to monitor fine-scale changes in dune morphology and demonstrated that “survey grade” data can be generated using such a technique, which otherwise is a popular photography technique for hobbyist aerial photographers. Two variations of the same foil (HQ KAP foil 1.6 $m^{2}$ and HQ KAP foil 5 $m^{2}$) were used to capture aerial photographs of the dune study system over six surveys between 30 March 2016 and 12 January 2018 to monitor structural changes on a fine spatial scale on an intra-annual and interannual scale.

The images were processed in Agisoft Metashape following the pre-processing of the KAP images to generate point clouds. For quantifying the changes in the dune topography between the surveys, the point clouds generated of the site were compared using a modified M3C2 approach [64,109] that included estimates for the point cloud to incorporate the variation in data captured over the KAP surveys. This was performed using the M3C2-PM plugin in CloudCompare. The KAP orthomosaics were found to have a spatial resolution of approximately 6 mm and point clouds of accuracy (x/y:19.7 mm and z:39.4 mm) that the authors state are far superior numbers than airborne lidar. Time series analysis of the inter- and intra-annual KAP data demonstrated that accretion and erosion at the scale of <500 mm could be quantified, which were highlighted as significant by the M3C2 analysis given the uncertainties in measurement, thus showing that this method is suitable for tracking fine-scale structural changes in coastal dune environments.

### 3.3. Terrestrial and Aerial Photogrammetry

Ref. [110] employed an approach optimising the complimentary properties of a VMS and UAV to monitor shoreline changes and intertidal topography to quantify erosion rates. This utilised the high temporal frequency of the video images and the high-resolution DEMs and orthophotos from UAV. The authors compared the outputs from these two methods and their complementarities. The shoreline of a particular cross-shore profile from timex images stacked to form an optimised image was delineated, optimising the changing R (red) to G (green) ratio of image pixels at the transition between land and sea. Aerial photographs captured from a quadcopter were subjected to an optimising SfM-MVS workflow to generate DEMs and orthophotos in Agisoft Metashape. The shorelines from these orthophotos were manually delineated in ArcGIS. A comparison between the monthly VMS and UAV shoreline locations found a correlation of 0.73 with a R^2^ of 0.50. The authors found that the VMS shoreline recession rates were overestimated, and shoreline advance rates were underestimated when compared against the UAV data, and they attributed these biases to various sources of errors such as water level variation and camera height. Even though VMS shoreline change rates were within the same range as that of the UAV shoreline change rate, higher deviations from the ideal regression line were observed during erosive periods than accretion periods. The intertidal profile was more accurate for the UAV data (vertical RMSE = 0.05 m) than the VMS due to the systematic error in the measurement of shoreline positions, which were subsequently used to derive the intertidal profile.

Ref. [42] mentions the utilisation of aerial photogrammetry using UAVs for topographic surveys and low-cost video monitoring stations using Raspberry Pi for shoreline monitoring and wave run-up within the STIMARE project (mentioned in Section 3.1.1). The authors mentioned using TLS surveys as a reference for validating the UAV derived topography. However, no validation intervention was implemented for the Raspberry Pi VMS.

**Table 4 sensors-23-01717-t004:** Summary of papers on aerial photogrammetry that includes type of paper, aerial platform/sensors used, parameters derived, software used to extract parameters, hazards/forcing agents monitored, validation, and accuracy.

Authors	Journal/Conference/Book Chapter	Aerial Platform/Sensors	Variables Derived	Software Used to Extract Variabless	Hazard/Coastal Zone Characteristics/Forcing Agent Monitored	Validation	Accuracy
[42]	Conference	UAV and VMS with Raspberry Pi	Topography, shoreline position, and wave run-up	Not mentioned	Storm surge	UAV validation with TLS. Not mentioned for the VMS	Not mentioned
[67]	Journal	Precision Hawk’s Lancaster Rev 3 fixed wing/RGB: Converted Nikon J3 14.2 MP and 3Drobotics Iris + Mapper VTOL quadcopter/RGB: canon S110 12 MP	DSM	PIX4Dmapper	Cliff/bluff morphological change	Check points using NRTK-GPS	Total average difference was used; fixed-wing DSM: −0.117 m; quadcopter DSM: −0.0224 m
[82,83,84]	Review papers						
[86]	Journal	Skywalker X8 flying wing (platform)/Sony Nex-5 R RGB camera (sensor onboard)	DSM	PIX4Dmapper	Storm-induced beach erosion	Ground control points measured with RTK-GPS, LoD (using a known reference area)	RMSE for the GCPs = 2.34–3.26 cm, S.d. using the reference area = 3.74 cm
[87]	Journal	FV-8 Atyges octocopter/24 Mpix Sony Alpha 7 full-frame sensor RGB camera	DEM	PIX4Dmapper	Storm-induced beach erosion	Individual checkpoints (ICPs) using DGPS	RMSE 6.89 cm for pre-storm DEM and 5.54 cm for post-storm DEM
[88]	Journal	DJI Phantom 2 (DP2) quadcopter/GoPro Hero 4 Black and DJI Phantom 4 Pro (DP4P)/20 Mpix camera	DSM	Agisoft Metashape and MATLAB	Beach-dune morphological change	Validation points using DGPS	RMSE and bias: for DP2 (0.13 m, −0.1 m, respectively); for DP4P (0.05, −0.02, respectively)
[89]	Journal	DJI Phantom 4 quadcopter/built-in camera FC330 and a ½.3″ CMOS sensor (12.4 Mpixel resolution)	DSM	Agisoft Metashape	Beach-dune morphological change	ICPs using GNSS	$R^{2}>0.98$ and RMSE = 0.173
[90]	Journal	DJI Phantom 4 Pro/CMOS sensor acquiring 20 Mpix RGB images	DSM	PIX4Dmapper	Beach erosion	ICPS using RTK-GPS	Normalised median absolute deviation (nmad) = 0.048 m–0.054 m. Mean errors and standard deviation; for 2018 ICPs (−0.044 m, 0.077 m, respectively) and for 2019 (0.128 m, 0.063 m)
[91]	Conference	DJI Phantom 3 pro/12 Mpix RGB camera	Shoreline position	Agisoft Metashape	Shoreline change	Not mentioned	Not mentioned
[92]	Book chapter	Aibotix Aibot X6V2/LiveMos 16 Mpix camera	DSM	Agisoft Metashape	Shoreline change	Control points (CPs) using GNSS	RMSE = 0.036 m
[93]	Journal	DJI Phantom 3 advanced quadcopter/Sony EXMOR 12.4 Mpix RGB camera	DSM	PIX4Dmapper	Dune morphological changes	GCP measured using RTK-GNSS followed by LOOCV	Mean error = −3 cm, RMSE = 8 cm
[94]	Journal	DJI Phantom 4 Pro/1″CMOS 20 Mpix RGB images	DEM	Agisoft Metashape	Sand dune migration and volume change	TLS	RMSE = 0.08 m, MAE = 0.06 m $R^{2}$ = 0.999
[95]	Journal	DJI Inspire 2 UAV/Zenmuse X7 camera	DEM	PIX4Dmapper	Coastal erosion	Control points using RTK-GNSS and TLS	RMSE and root sum of squared errors (RSSE) from the control points = 0.040 m and 0.046 m, respectively. Mean error and RMSE from the TLS = 0.02 m and 0.04 m, respectively.
[96]	Book chapter	Not mentioned	DEM	Agisoft Metashape	Sandy beach erosion	Validation points using DGPS	RMSE 0.95–30 cm
[108]	Journal	HQ KAP foil 1.6 and 5 m^2^/Canon D30 compact digital camera	3D point cloud	Agisoft Metashape	Dune erosion	GCPs measured using DGNSS	RMSE = 27.9 mm
[110]	Journal	Not mentioned	DEM/shoreline position	Not mentioned	Beach erosion/intertidal topography	GCPS using RTK-DGPS	RMSE for the VMS derived DEM = 1.4 m–4.6 m.Mean error for the UAV derived DEM = 0.25 m.

### 3.4. Global Navigation Satellite System Reflectometry (GNSS-R)

The GNSS consists of four constellations of navigation satellites, namely GPS, GLONASS, BEIDOU, and GALILEO, maintained by different countries. The reflected signals from these satellites could be used for different environmental monitoring such as coastal sea levels as demonstrated by [111,112,113]. These low-cost GNSS receivers, unlike conventional tide gauges, need not be inserted into the water body for the monitoring of tidal levels, which could corrode the instrument and are not as expensive as the radar gauges.

While measuring the relative sea level rise for a region, any vertical land motion (VLM) must also be considered to obtain the correct measure for the local sea level rise. Conventional tide gauges do not take this into account and hence are often co-located with GNSS antennas. Low-cost GNSS-Reflectometry accounts for this VLM. There are different approaches for GNSS-R measurement [114].

Ref. [113] utilised a ground-based phased altimetry approach for the GNSS-R measurements, where a GPS zenith facing antenna receives the direct signal from the satellite that is right-hand circularly polarised (RHCP) and a nadir-facing antenna collects the reflected signal, which undergoes a change in polarisation to become the left-hand circularly polarised (LHCP) light. The reflected signal undergoes a path delay as compared to the direct signal and this delay is the basis of the phase altimetry approach. The observed carrier phase of the direct and reflected signals can be calculated using the I and Q observation output of the master channel and the slave channel of the receiver [113,114]. Ref. [113] validated their measurements of sea surface level height against pneumatic tide gauges and found that the measurements were highly correlated with a correlation coefficient (r) of 0.93 and a RMSE of 4.37 between the GNSS-R and the tide gauge measurements. The antennas used were off-the shelf and cost about USD 300.

Ref. [111] developed an open-source Arduino platform for SNR based GNSS-R costing about USD 200 for the measurements of water level, which on comparison to a co-located tide gauge resulted in a correlation coefficient of 0.989 and RMSE of 2.9 cm. This experiment was carried out at a lake; thus, the results demonstrated a best-case scenario, and its results are yet to be seen in the open ocean, nonetheless showing promise. Similarly, [112] utilised SNR based GNSS-IR to measure tidal water levels, and the measurements were validated against co-located tide gauge data. This paper utilised the cheapest GPS hardware that costs USD 30. The GPS system was involved with the objective of providing better information on tidal levels, especially during the stormy winter season in Ireland, when inaccuracies in tidal predictions could be a threat to civilian safety commuting to a nearby island via causeway. The outputs, when compared to a co-located bubbler tide gauge, were found to be in excellent agreement, with a RMSE of 1.7 cm for daily averages and 5.7 cm for tidal range, exceeding 3 m at spring tides. GNSS-R optimised low-cost sensors were seen to perform as good as, if not better than, the conventional instruments; the reason for this cannot be clearly attributed to any one factor [112] but seems promising to be deployed in places where installing expensive sensors could be cost-prohibitive.

### 3.5. Wireless Sensor Network

Wireless sensor networks have been used for low-cost monitoring of several coastal forcing agents/hazards [115,116,117,118,119] in often hostile or inaccessible environments, facilitating the study of several fundamental processes that otherwise would be rarely studied due to their inaccessibility [120].

The vulnerability of coastal populations to increasing risk due to storm surges is increasing and the damage incurred by the same cannot be overlooked. Ref. [117] developed a prototype low-cost wireless system for multipoint storm surge measurement with the objective of providing near-real-time information on storm surges to develop robust prediction models based on near-real-time, high-resolution measurements of storm surges on a local scale. Furthermore, the authors also emphasise that a data-driven model taking in inputs of real-time measurements of water level and meteorological data from such a prototype wireless system would be ground-breaking. There has been work around data-driven statistical models for predicting storm surges based on a set of predictors [121], but none use real-time storm surge information. The conventional instruments for storm surge measurement are either too expensive, (running in thousands of USD) or are unable to transmit data in real time, only providing data after the storm has passed [117]. To mitigate these drawbacks, the authors designed a wireless sensor unit (called the sensor head) consisting of various components such as a water pressure sensor, microcontroller, and various other electronic components.

The operation of this prototype was implemented within three subsystems. The first subsystem is the field installation, which is responsible for sensing, data acquisition, and data management. The second is the communication layer; this is the private wireless network that sends the data collected by the sensor head to a local laptop installed in the field via Wi-Fi or cellular data networks using the IEEE 802.15.4 protocol. The laptop then passes on this data to an optimising server at a remote location, thereby rendering the real-time feature to this low-cost sensor. Even in the instance of failure of this real-time link, the chances of data loss are negligible, as it is stored locally on the PCB mounted SD card as well as the local laptop. It is also possible to send the data directly to the central server without the need of a local laptop. The third subsystem is the central server or cloud storage, which processes the received data in near-real-time and presents the data to the end users in various ways such as graphs and other software applications.

The water level outputs from this sensor were compared against co-located NOAA tide gauges, showing good agreement. This prototype, in addition to being low-cost and having low-power usage, is portable, allows near-real-time optimising of storm surge data, has multiple modes of operation, and is versatile, as it could be utilised to measure other environmental parameters as well.

Refs. [115,119] implemented a wireless video sensor network (VSN) in conjunction with an instrument scheduler based on a real-time clock for energy-neutral monitoring of coastal erosion. Wireless sensor networks such as video sensor networks have made possible the study of complex processes in remote locations. The remote location of the coastal sites could often mean that a tethered energy infrastructure is not available at those sites, creating the need for optimised energy harvesting and power management strategies [115,120]. The objective of this instrument scheduler was to schedule the duty cycle of the VSN in such a way so that the sensing took place during the targeted periodic events; otherwise, the sensor was turned off for energy conservation. The VSN consisted of a camera network and wireless antennas that were arranged into camera nodes, relay nodes, and a base station for wireless data acquisition and transmission to a remote server. This VSN facilitated the identification of erosion and its forcing factors.

Ref. [116] developed a wireless sensor network (WSN) for the real-time monitoring of sand height variation in sandy beaches and dunes to quantify coastal erosion. An ad hoc sensing pole was developed, formed by an array of 24 light-dependent resistors or LDRs positioned 5 cm from each other to measure sand level variation with an accuracy of 5 cm based on the principle that the resistance of an LDR is inversely proportional to the intensity of light falling on it. Thus, assuming that half of the sensing pole is submerged in the sand and the other half is exposed above the ground, the 12 exposed LDRs would return a higher value of the current and the other submerged 12 would return a lower value. So, counting the number of LDRs (identified by low values) could be used to measure the height of the sand layer. The authors took necessary protections to secure the sensing poles from the harsh marine environment. Besides the sensing pole made of LDRs that was the principal hardware, there were other components such as analog multiplexers (MUX) and MOSFET transistors that were used in the construction of the “sensors nodes”. The sensor node consisted of three parts: the sensing pole with the array of LDRs carrying out the physical measurements; a logic part; and the wireless data transmission part consisting of the Xbee radio module. This system was not evaluated against any standard reference instrument, for instance TLS, and was instead manually validated on site by counting the number of submerged LDRs.

Ref. [118] describes a WSN-based coastal observation system called “The Coastal Ocean Observation System of Murcia Region” or OOCMUR to study climate-induced changes in oceanographic and ecological processes in the hypersaline coastal lagoon of Mar Menor, located in SE Spain. This WSN consists of four buoy-based sensor nodes, namely the depth node that takes samples of sea level and temperature; the current meter node that measures current speed and direction; “full nodes” that measure sea level, current, salinity, and temperature; and the watercourse node that measures ecological and water quality parameters as well as sea level, temperature, and salinity. These nodes are spread across strategic locations within the lagoon. Both GPRS and Zigbee modules are used for wireless communication. A LabVIEW [122] user application was implemented for the harvesting, processing, and storage of information from the sensor network. The data from the trial deployment of this WSN were validated against data from the Spanish Meteorological Agency (AEMET), located 15 km away from the location of the sensor, and exhibited good correlation, as seen in Table 5.

### 3.6. GPS Buoys

GPS buoys have several advantages over the conventional wave buoys fitted with accelerometer-tilt-compass sensor package due to their low-cost and reduced bulkiness [26,123,124,125].

To test the efficacy of the off-the-shelf satellite-based augmentation system (SBAS) enabled GPS receivers, Ref. [123] tested several such GPS receivers against conventional motion sensor packages and optimising GPS sensor packages to explore the utilisation of such low-cost GPS motion sensor packages in a low-cost drifter. Low-cost drifters have been explored in many previous coastal studies [126,127,128]. The low-cost GPS receivers were connected to a fleet of Datawell Directional Waverider (DWR) buoys and measurements were recorded from the low-cost GPS sensors and the internal buoy sensor packages. The fleet of DWR buoys consisted of buoys with a conventional motion sensor package as well as newer buoys consisting of optimising GPS sensor packages. All the GPS systems were configured to record north–west–vertical position data continuously with a sample rate of 1 Hz. Three GPS systems were evaluated; Magellan mobile mapper CX, Locosys Genie GT-31, and GlobalSet MR-350. Except for the Magellan mobile mapper CX, the remaining two demonstrated bias in the vertical displacement measurements of the buoy, thereby skewing the spectral wave statistics (wave frequency spectrum, mean wave direction, and directional spread) derived from the vertical displacement spectrum and first-order directional moments. However, the same spectral wave statistics for all the three GPS systems derived from the horizontal displacement spectra and second-order directional moments demonstrated excellent agreement with the all-internal buoy sensor packages (accelerometer–tilt–compass and Doppler shift in GPS measurement sensors). Thus, reliable routine wave information can be extracted from such GPS systems. The Magellan mobile mapper CX yielded satisfactory output for both the vertical and horizontal displacement spectra and thus was found to have the full capabilities of a heave–pitch–roll wave measurement system. Estimates of wave energy and direction spectra from the prototype drifters fitted with Magellan were found to be in good agreement with the Datawell buoys except at the high frequency spectral tail. Overall, these low-cost sensors could be utilised for accurate and routine wave monitoring. Even though satellite altimetry can provide accurate wave height estimates, it does not provide spectral wave information, which could be achieved by these prototype drifters fitted with low-cost GPS sensors.

The directional wave spectrum describes the irregular and unpredictable sea surface in the presence of wind-generated waves [129]. Such a directional wave spectrum is essential not only for wave modelling purposes but also to quantify the consequence of interaction of such waves with matter, for instance, in wave-induced erosion [129]. Ref. [124] describes a low-cost GPS based wave buoy called the directional wave spectra drifter (DWSD) developed by the Lagrangian Drifter Laboratory (LDL). The outputs from this sensor were compared with measurements from a co-located bottom-mounted acoustic doppler current profiler (ADCP) and it was found that this prototype drifter responded better to high-wave frequencies, which can potentially mitigate the problem of the low-cost drifters fitted with Magellan CX as described by [123]. The DWSD is being explored to constrain the wave energy climatology for optimising the development of a full-scale prototype wave energy converter in the gulf of Naples, Italy. To quantify the difference in the wave measurements obtained with the two instruments, statistical metrics such as bias and root mean square error (RMSE) were used, and it was found that the bias and RMSE of the main wave parameters were well within the acceptable range for most coastal engineering applications (see Table 5).

Ref. [26] developed a low cost GNSS-buoy platform to measure coastal (local) sea levels. The buoy platform was constructed from off-the-shelf components. The low-cost single frequency receiver called the U-blox M8T GNSS receiver [130] was used for measuring the sea levels using a post-processed kinematic (PPK) solution in RTKLIB open-source software [131]. The UK ordnance survey (OS) geodetic GNSS site was used as the base station for the PPK solutions, with a baseline of 200 m. This low-cost buoy was attached to a mooring buoy and allowed to float freely at a water depth of 4 m below the chart datum. Data recorded from this buoy were validated against the co-located Environmental Agency (EA) tide gauge data. A Van de Casteele test [132] was used to test the accuracy of the GNSS-buoy and the resulting Van de Casteele diagram shows a good agreement between the gauges, with a mean difference = −0.011 m, standard deviation = 0.009 m, and RMSE = 0.014 m. Additionally, fingerprints of harbour oscillations were found in the GNSS data.

Ref. [125] developed a low-cost GNSS buoy with a self-assembled inertial measurement unit (IMU) for accurate positioning data in situations when the GNSS signal might be attenuated. A lifebuoy was used as the platform for the assembled hardware, which consisted of a dual-frequency geodetic-grade Trimble R4 GNSS receiver. Validation was carried out with a standard reference wave gauge in the laboratory as well as in the field. Though the validation experiments returned favourable outputs for the IMU integrated GNSS buoy, the authors did not mention any statistical metric to quantify the accuracy of the buoy. Furthermore, the advantage of integrating IMU on GNSS accuracy could not be quantified, as the buoy was deployed in the inner harbour without harsh ocean environmental conditions.

### 3.7. DIY Pressure Sensor/Gauge

This section and the following Section 3.8 review measurement of water levels using pressure sensors to discern information on waves, tides, and storm surges. The governing equation is as follows [133]:(6)$$d=\frac{\mathrm{AP}-\mathrm{BP}}{g\rho }$$

Here d is the water depth, AP is the absolute pressure, BP is the barometric pressure, g is the acceleration due to gravity, and ρ is the density of sea water.

Do-it-yourself (DIY) pressure sensors are an alternative to expensive commercial wave gauges [134,135]. Refs. [134,135], constructed DIY open-source Arduino-based wave sensors or gauges using the digital pressure sensor, MS5803-14BA. Commercial plumbing hardware was used to build the gauge. The pressure sensor, which is left exposed, measures water levels and the sampling is controlled through codes written in the Arduino IDE. The performances of these wave gauges were validated against commercial wave gauges; while [134] carried out the validation both in the field and laboratory setting, Ref. [135] conducted the validation only in the field. The laboratory testing was performed to get rid of any confounding variables to better optimise factors that could have an influence on the sensor output. For the field testing, Ref. [134] deployed both the DIY wave gauge and the commercial wave gauge to a depth of 1 m at high tide and suitably mounted these to a fixed support. Initial processing of the DIY pressure data revealed missing data (<1%) that were estimated using linear interpolation in MATLAB. All statistical analysis, such as paired t-test and linear regression modelling to quantify the agreement between the DIY and commercial gauge for the laboratory test, was carried out in R. The t-tests revealed no significant difference (*p* ≥ 0.7) and the R^2^ values ranged from 0.69 to 0.91, the lower values being recorded for the wave tests having high-frequency waves. Field test data were compared not only in terms of linear regression between the DIY and commercial gauge pressure data in R, but also in terms of spectral analysis in MATLAB. Wave energy density distribution had 92% agreement between both the gauges and model fit was excellent, with R^2^ = 0.997. The validation of the wave gauge developed by [135], also called the open wave height logger (OWHL), with a co-located commercial wave and tide recorder exhibited a very high statistically significant correlation (r > 0.99, *p* < 0.0001) for the sea surface elevations, wave height, and period. Furthermore, the OWHL’s significant wave height and period were compared against two nearby Datawell Waverider MkIII surface buoys and two other commercial wave buoys (CDIP buoy 028 and CDIP buoy 092) located a few kilometres (22.09 km NW and 17.6 km SE) from the test site. Comparison with the nearby Waverider buoys found generally strong correspondence depending on the distance between the sensors and ocean swell conditions. As the CDIP buoys were located several km offshore, the significant wave heights reported by them were generally larger than the OWHL estimates. Similarly, even though the sensors were able to capture the shift between short-period wind waves and long-period ground swell, there were offsets at certain times that could be driven by several confounding factors such as local bathymetry and sensor type. A second validation run was conducted where the OWHL was bottom-mounted beneath a Waverider MkIII surface buoy (CDIP buoy 158) as a direct reference to compare wave data. The correspondence between the significant wave height and wave period was good, but better for the wave height (r = 0.985, *p* < 0.001) than the wave period (R = 0.736, *p* < 0.001).

The sampling rate of these two DIY wave gauges [134,135] were 8–10 Hz and 4 Hz respectively, which implies that the characterization of high frequency waves might be limited, and even though the Arduino code could be modified to sample at a faster rate, there are practical limits set due to factors such as attenuation of pressure with increasing depth, which requires pressure attenuation correction to the pressure signal from the low-cost digital pressure sensor [135,136]. Ref. [134] also found a lower value of R^2^ for laboratory wave tests having high frequency waves. Thus, Ref. [135] suggests that for studies primarily focusing on measuring high frequency waves, other sensing methods might be more appropriate.

### 3.8. Water Level Sensor

Refs. [133,137] utilised low-cost commercial water level sensors (pressure sensors) to measure estuarine storm surges in Maine, USA and nearshore wave morphology and tides along Wallops Island, Virginia, USA to ultimately contribute towards accurate coastal flood modelling by filling the data gaps in local areas often lacking suitable data for calibrating and testing numerical coastal storm surge models.

Ref. [133] employed a citizen science approach for sensor deployment and data collection to measure estuarine storm surges in four Maine estuaries using a network of low-cost commercial water level sensors. A citizen science kit consisting of a water logger, a water shuttle, concrete mooring, and computer software were imparted to the participants along with relevant training. There were 22 stations collecting absolute pressure data and 3 stations collecting sea level pressure data to barometrically correct the water level. Additionally, 15 of these 22 stations also had a control station where data was collected by a University of Maine researcher for data quality check of citizen-science-collected water levels.

Storm surge was calculated from the citizen-collected total water levels after checking for data completion, accuracy, and contamination in the data processing stage. The storm surge data helped address research questions such as the impact of estuarine morphology and tide–river interaction on low-frequency storm surges [138], demonstrating the reliability of such data for publication in peer-reviewed journals. Due to lack of sufficient water level data in the estuaries, the data gathered through this approach were also useful for validating operational storm surge models to identify gaps to improve forecasts. There was, however, no discussion on the validation of the data against a relevant high-cost reference instrument, possibly because the sensors used were industry standard and the reference instruments were too far apart to result in any viable comparison.

IoT enabled ultrasonic water level sensors are also being used for effective flood monitoring and forecasting within the StormSense project [139,140]. The StormSense project uses IoT bridge-mounted ultrasonic sensors that transmit data via long-range wireless area networks (LoRaWAN) and integrates several extant water level sensors from NOAA and USGS and a tide gauge operated by the Virginia Institute of Marine Science (VIMS). Some of the main objectives the project addresses are: archiving water level observations for flood reporting, automating targeted advance flood alert messaging, and calibration/validation for hydrodynamic flood models [141]. The StormSense ultrasonic sensors were co-located with the USGS radar sensors temporarily for validation and accuracy measurements, with an accuracy of ± 18 mm found in the field measurements. Measurements in the lab revealed a RMSE of ±5 mm. This elucidates the significant differences that could result in field and lab measurements.

Ref. [137] used low-cost “ground water sondes” to collect local data to measure nearshore waves (where buoy-deployed water level sensors are impractical) and tides to better predict coastal flooding and shoreline change by using such data for calibration and validation of coastal numerical models. These groundwater sondes are so called because these are used to monitor local groundwater hydrology and water quality [142]. The study was carried out in Wallops Island, Virginia, where information on the local tides and wave conditions was scarce, reducing the effectiveness of numerical models to predict shoreline change and flooding along this shoreline, which commonly faces flooding and erosion in times of elevated water levels. The tidal data collected from two sites, few kilometres apart, exhibited almost perfect overlap (no metric was provided to quantify the correlation, only a graph was provided), with a lag of 60 min in the arrival time of the tides between those sites. Furthermore, comparison with a NOAA tide gauge located 80 km from Wallops Island shows significant differences in magnitude and timing. Such observations clearly expose the inconsistencies in the current tidal predictions for the region, as the tidal timing and range are affected by positions within the inlet and back bay, further highlighting the importance of local measurements. For nearshore wave measurements, three sensors were attached to a 30 m long array installed on the seafloor in two locations along Wallops Island. The measurements were able to resolve the wave morphology accurately. 

### 3.9. Ground-Based Beach Profiler

Refs. [143,144,145] used three different techniques for surveying beach profiles for the quantification of beach morphological changes such as beach erosion [146]. Ref. [143] carried out the beach profiling using a Topcon total station along a low wave-energy coast in west-central Florida to quantify the changes in beach profile post-nourishment (artificial addition of sand) of a few of the critically eroding barrier islands located in that region as well as post-storm impacts focusing on the period 2012–2015. The authors successfully demonstrated the efficiency of the technique in quantifying shoreline changes and bar migration due to beach nourishment and tropical storms, respectively, which not only provided quantitative evaluation of the beach restoration works for optimising beach nourishment design, but also provided insight into coastal management in general. The authors argue that even though this technique is labour-intensive, it is much cheaper than using airborne/terrestrial lidar, allowing for greater temporal coverage. However, there was no comparison with an equivalent lidar dataset. A comparative study was made by [147] of four different surveying techniques using a total station (a reference instrument for measuring location using GPS) and a RTK-GPS and it was found that both were highly accurate and adequate for monitoring morphological changes at a small area of the beach, further emphasizing the adequacy of the total station.

Ref. [145] designed a wireless beach profiler (WBP) and compared the outputs to an electronic distance meter and the outputs obtained via the Emery method [148]. The main equipment of the WBP is an accelerometer for measuring the tilt along several points in a path perpendicular to the coastline. The tilt is proportional to the beach slope, which leads to the construction of a beach profile along a particular transect. Each measured point was in polar coordinates (β, d), where β is the angle of tilt and d the distance between two consecutive points. The output of this device is dependent on the local temperature, necessitating a digital temperature sensor to compensate for the output. The WBP also consisted of a GPS receiver, a microcontroller, a RF transmitter, a RF receiver, and a handheld PDA (personal digital assistant). The validation of the WBP with the EDM and Emery method provided statistically insignificant differences (*p* > 0.05). The WBP was automated with the help of an autonomous robotic vehicle (AWBP) and the results were compared to the Emery method and manually operated WBP to yield statistically insignificant differences.

Ref. [144] designed a land-based platform for sandy shore monitoring (volumetric quantification of sediments and shoreline delineation) called INSHORE (Integrated System for High Operational Resolution in shore monitoring). This system was evaluated in a reflective intermediate sandy shore in Vagueira, Portugal. The INSHORE system determines the GPS ground coordinates of the surveyed area with high vertical accuracy (1–2 cm) by attaching GPS receivers and a laser distance sensor to a sandy-beach-suited vehicle such as a motor quad [149]. The GPS coordinates, besides being used for shoreline delineation, can be used for generating a DEM by using interpolation procedures on the coordinates, allowing the determination of beach volumes. This is possible due to the collection of several points in cross-shore and along-shore transect by the INSHORE, yielding sufficient data points for the generation of a DEM. A triangular-shaped structure was fitted onto a motor-quad such that two vertices were fixed along a side of the motor quad, while the third was horizontally out of the vehicle. On the two inner vertices were two low-grade GPS receivers connected to an L1 GPS antenna, and a high-grade GPS receiver was attached to the third vertex connected to dual-frequency GPS antenna. A fourth GPS receiver was installed over a fixed point near the survey site for DGPS processing. Below the dual frequency antenna is a laser distance sensor to measure the vertical/slope distance to the ground.

Four validation tests were conducted by establishing a test grid of 20 control points set up with DGPS. The first two surveys were conducted in the static mode: grid surveying in the static mode with roll and pitch angle variations, respectively; and the last two were conducted in kinematic mode: grid surveying in kinematic mode with reduced and moderate velocity, respectively. For accuracy measurements, refer to Table 5.

### 3.10. High Wind Speed Recording System

Ref. [150] developed an energy-efficient and low-maintenance high wind speed recording system (HWSR) for the continuous monitoring of wind speed and direction, especially during depressions and cyclonic storms along the eastern and western coasts of India when the probability of power outages is high and could lead to the failure of the existing power-intensive wind monitoring instruments. The main components of the HWSR are the potentiometric wind vane for the measurement of the wind direction and an optical anemometer for measuring the wind speed. The instrument can operate continuously for 25 days on a 60 AH battery without main power supply. A solar panel trickle charges the battery, ensuring years of uninterrupted operation. The system has a data storage capacity of 10 years for one-minute averaged data of wind speed and direction, which can be retrieved through a USB port. For validation, the outputs of the HWSR were compared to the wind data from the conventional observatories, showing good agreement.

### 3.11. UAV RTK-Lidar System

Unlike the UAVs being used for photogrammetry as discussed in a previous section, Ref. [151] used this airborne platform for wave and tide measurements. An RTK lidar system consisting of a robotic scanning lidar, an altitude and heading reference system (AHRS), a RTK GNSS and a small i7 industrial PC was integrated into a multirotor UAV weighing just under 11 kg. The temporal resolution of the lidar, AHRS, and the RTK systems were 40 Hz, 100 Hz, and 20 Hz respectively. All the data were recorded on the PC with time stamps. All data were resampled in a time frame of 20 Hz with linear interpolation. Validation of this system was carried out in an inter-tidal zone near YongAn Harbour, Northern Taiwan. A strain gauge pressure sensor equipped in an acoustic doppler velocimetry (SonTek ADV-Oceans) was used to validate the results measured by the UAV system. The RMS errors for the tidal elevation, significant wave height, and wave period measurements between the two techniques were 4.9 cm, 4.8 cm, and 0.028 s, respectively, showing good agreement. While the temporal resolution was fixed at 20 Hz (after resampling), the spatial resolution ranged from 0.4 cm to 8.7 cm for flight heights ranging from 1 to 20 m, therefore constraining the flight height to much lower altitudes for useful return data rate. It is also suggested by [151] that by combining this arrangement with UAV imagery techniques, it is possible to observe land and sea surface signatures in one flight measurement.

### 3.12. Cable-Mounted Robot for Near Shore Monitoring

Ref. [152] developed a prototype robotic monitoring platform to complement the existing monitoring networks. The robotic system can traverse a partially submerged cable that has one end attached under water and the other end attached onshore. When the robot is deployed, it moves along the cable and submerges itself to a target depth or position and uses on-board sensors to take a measurement or series of measurements before returning to the surface. In this way, it can collect data repeatedly from a single position while also resurfacing to transmit data to a remote site. The current prototype has an operating depth of 3 m and is intended to complement several pre-existing oceanic observation platforms and sensor suites, such as buoys, moorings, and HF radar systems. Details of the sensor components can be found in [152] and can be roughly classified into two main assemblies: the drive assembly that carries the robot along the cable and the control housing, which holds the electronics and sensors. The authors also state several benefits of the traversal mechanism, such as the ability to mount the robot in locations with sharp cliffs or rough and rocky water where sustained human presence is difficult or dangerous. Work is underway to further improve the design of this prototype to develop it as a modular commercial product. Though the authors do not present any data acquired from the system, they see its potential in monitoring extreme weather events along the coastline. One of the sensors that they used on this platform is the Aqua TROLL 500 sonde, which is a multiparameter sonde capable of measuring pressure, conductivity, and temperature. The data collected from this sonde attached to the platform are presented in [153], where these data were validated against human-collected data.

**Table 5 sensors-23-01717-t005:** Summary of papers on GNSS-R, WSN, GPS-Buoy, DIY pressure sensor/gauge, water level sensor, ground-based beach profiler, high wind speed recording system, UAV-RTK Lidar system, and cable-mounted robot for nearshore monitoring, which includes: type of paper, principal sensing components, parameters derived, software used to extract parameters, hazards/forcing agents monitored, validation, and accuracy.

Authors	Journal/Conference	Principal Sensing Components Used	Variables(s) Derived	Software(s) Used to Extract Variables	Hazard/Coastal zone Characteristics/Forcing Agent Monitored	Validation	Accuracy
[26]	Journal	U-blox M8T GNSS receiver	Coastal water level	RTKLIB open-source software	Sea level rise	Tide gauge	Difference = −0.011 m, standard deviation = 0.009 m, and RMSE = 0.014 m
[111]	Journal	Arduino-based sensor with a single-frequency GPS L1 C/A add-on (Adafruit GPS FeatherWing); and an external GPS patch antenna (28- dB active, Chang Hong GPS-01-174-1M-0102)	Sea level	Arduino IDE and MATLAB	Sea level	Radar gauge	r = 0.989 and RMSE = 2.9 cm
[112]	Journal	Maestro A2200A SiRFstar IV module	Tidal water level	Not mentioned	Tidal water level	Tide gauge	RMSE of 1.7 cm for daily averages and 5.7 cm for tidal range exceeding 3 m at spring tides
[113]	Journal	GNSS occultation, reflectometry, and scatterometry (GORS) receiver; Antcom L1/L2 dual frequency antenna	Sea level	Not mentioned	Sea level	Tide gauge	r = 0.93, RMSE = 4.37 cm
[115]	Conference	Same VSN as [121] with the added advantage of an Arduino-based instrument scheduler	No quantitative parameter derived	Not mentioned	Coastal erosion and the corresponding forcing agents such as tides, waves, wind	Not mentioned	Not mentioned
[116]	Journal	Sensor node consisting of many different electronic components with the principal sensors being the LDR	Sand level variation	Not mentioned	Coastal erosion	Manual validation	Not mentioned (just qualitative “good accuracy”)
[117]	Journal	Sensor unit consisting of many electronic components with the principal sensor being the water pressure sensor	Coastal water level	MATLAB, ThingSpeak	Storm surge	NOAA tide gauges	Not mentioned
[118]	Journal	Sensor node consisting of various sensors such as water pressure sensor, temperature and salinity probe	Multiple parameters, but the parameter of interest within this review is coastal water level	LabVIEW	Sea level rise	Atmospheric data from the Spanish meteorological agency	Squared coherence of 0.85
[119]	Journal	A VSN with a camera node made up of AXIS M1101 network camera and a PicoStation2 antenna	No quantitative parameter derived	Not mentioned	Coastal erosion and the corresponding forcing agents such as tides, winds, waves	Not mentioned	Not mentioned
[123]	Journal	GPS receivers; Magellan mobile mapper CX, Locosys Genie GT-31 and GlobalSet MR-350	Wave parameters	Not mentioned	Waves	Wave rider Datawell buoys	Good accuracy except at high frequency
[124]	Conference	A GPS buoy called directional wave spectra drifter. There is no specific mention of the type of GPS receiver	Wave parameters	Not mentioned	Waves	ADCP	Bias and RMSE for significant wave height, mean wave period, peak wave period, and peak wave direction are (0.03 m, 0.05 m),(−0.02 s,0.2 s), (0.3 s,0.7 s), and (3.7°,9.9°), respectively
[125]	Conference	Geodetic-grade Trimble R4 GNSS receiver and an IMU unit	Wave parameters	Not mentioned	Waves	Reference wave gauge	Qualitative: “good agreement with the reference wave gauge only with increasing wave height”
[133]	Journal	HOBO U20 Water Level Logger and HOBO MicroStation	Coastal water level	MATLAB	Storm surge	Not mentioned	Not mentioned
[134]	Journal	Arduino-based pressure sensor MS5803-14BA	Coastal water level	Arduino IDE, MATLAB, and R	Waves	Commercial wave and tide gauge	Lab; p ≥ 0.7, R2 = 0.69 to 0.91Field; R2 = 0.997
[135]	Journal	Arduino-based pressure sensor MS5803-14BA	Coastal water level	Arduino IDE, and R	Waves	Commercial wave and tide gauge	r > 0.99, p < 0.0001
[137]	Journal	Solinst Levelogger LT Gold Series and a Barologger Gold pressure sensor	Coastal water level	Solinst Levelogger V3.4.0 Software	Waves and tides	Reference gauges located far away >80 km from the low-cost sensors	Not mentioned
[140]	Conference	Valarms IoT ultrasonic sensors	Coastal water level	Not mentioned	Storm surge	USGS radar sensors	Lab; RMSE = 5 mm Field; RMSE = 18 mm
[143]	Journal	Topcon total station	Beach profile	Not mentioned	Coastal erosion	Not mentioned	Not mentioned
[144]	Journal	1 L1 GPS antenna, 2 low-grade GPS receivers, 1 L2 GPS antenna, 1 high-grade GPS receiver	Shoreline position and DEM	Not mentioned	Coastal erosion	Test grid of control points measured with DGPS	Mean altimetric error was within 2 cm
[145]	Journal	Wireless beach profiler	Beach profile	Associated software of the PDA (Hewlett-Packard iPAQ hx2790, Palo Alto, California)	Coastal erosion	Electronic distance meter and the Emery method	No statistically significant difference (p > 0.05)
[150]	Journal	Potentiometric wind vane, optical anemometer, and various other electronic parts	Wind speed and direction	Not mentioned	Storms	Cup counter anemometer in the conventional observatories	Qualitative: “good agreement”
[151]	Journal	UAV: DJI, S1000, scanning lidar (Hokuyo, UTM-30LX), AHRS (Xsens Technologies, mTi 30), two GNSS receivers (NovAtel, OEM 628), two antennas, and two lightweight portable radios (433 MHz)	Tides and waves	Not mentioned	Tides and waves	Strain gauge pressure sensor equipped in an acoustic Doppler velocimetry (SonTek ADV-Oceans)	RMS error for the tidal elevation, significant wave height, and wave period measurements between the two techniques is 4.9 cm, 4.8 cm, and 0.028 s respectively
[152]	Conference	A robotic platform consisting of the Aqua TROLL 500 sonde, which is a multiparameter sonde	Multiple parameters, but the parameter of interest within the review is barometric pressure	Not mentioned	Storms	Human-collected data	% difference is 1.04

## 4. Discussion

This section discusses the key outcomes of this systematic literature review focused on the availability of multiple low-cost sensors to monitor similar (e.g., DEM and beach profile) as well as different variables (e.g., coastal water levels), then finally links those variables with the corresponding broad category of coastal hazards (coastal/storm surge flooding, coastal erosion, and shoreline change).

Coastal hazard monitoring includes, in addition to the monitoring of the hazards themselves, their forcing agents and their induced coastal responses, such as shoreline change and changes in coastal topography [10,22]. Figure 3 represents the physical characteristics, forcing agents, and hazards monitored by the sensors reviewed in this article. As seen from the figure below, within coastal zone characteristics, the low-cost sensors monitor the coastal topography of the beach-dune system, cliffs, and inter-tidal regions. The forcing agents that are also the key metocean variables responsible for the hazards such as coastal water level, wave characteristics, wave set-up, tides, and barometric pressure along with the hazards themselves are monitored. Though the variables within these three categories are not exhaustive, they are fundamental for climate-induced coastal hazard monitoring [18], refs. [10,22] to urgently monitor ungauged coastal regions highly vulnerable to the impacts of the changing climate.

Figure 4 below is a schematic of the different categories of low-cost sensors and the variables monitored. The variables are colour-coded to highlight that different monitoring techniques exist to monitor similar variables.

The key inference from this schematic is the availability of multiple low-cost sensing methods to monitor the same variable; for instance, GPS-Buoy, DIY pressure sensor/gauge, GNSS-R, and commercial low-cost water level sensors such as groundwater sondes could be used to measure coastal water levels, facilitating the researcher/coastal manager to not only choose from a wide range of options, but also adapt the sensing methods according to their specific needs. For instance, as seen from Section 3.6. GPS Buoy, certain components of the low-cost sensors such as GPS receivers used in GPS buoys or for GNSS-R measurements can be modified without necessarily having to stick with the same electronic modules used by the authors.

The variables in Figure 4 can be further simplified; for instance, DEM/DSM, point clouds, and sand level variation essentially quantify the volumetric changes in sediments [144] for monitoring coastal erosion, leading to another key point, that is, to link these variables to coastal hazards.

The variables in this review can be mapped onto three broad hazard categories as shown in Table 6: coastal flooding/storm surge flooding, coastal erosion, and shoreline changes. Although many authors use coastal erosion and shoreline changes interchangeably [10], they are different [13] and are treated separately here. Coastal erosion involves morphological changes, whereas shoreline change is a change in the shoreline position or shoreline proxies, which does not necessarily cause morphological change [13]. As seen from Table 6, several variables can be mapped to a single hazard; for instance, coastal water level, waves, wind, and coastal topography are important variables for monitoring coastal erosion.

Information on coastal topography derived from variables such as DEM/DSM, 3D point clouds, sand level variation, and beach profile helps in the quantification of volumetric changes in sediments quantifying accretion or erosion rates in beach-dune systems and hence have been categorised under coastal topography, as seen from the table below. DEM/DSM and 3D point clouds are derived from images taken either on ground [60,61,62,63] or from an aerial platform such as a drone [86,87,88,89,90,91,92,93,94,95,96] or a kite [108] fitted with an RGB camera.

**Table 6 sensors-23-01717-t006:** Climate-induced coastal hazards and the relevant variables monitored.

Variable(s) Hazard(s)	Coastal Topography (DEM/DSM, 3D Point Cloud/Beach Profile/Sand Level Variation)	Shoreline Position/Shoreline Proxies/Waterline	Coastal Water Level	Tides	Wave Run-up/Wave Characteristics	Wind Direction/Speed	Barometric Pressure
Coastal flooding/storm surge flooding		√	√	√	√	√	√
Coastal erosion	√	√	√	√	√	√	
Shoreline change	√	√	√	√	√	√	

The obtained DSMs and point clouds from processing images via structure from motion (SfM) algorithms [56] implemented in photogrammetric software such as Agisoft Metashape, PIX4dmapper, etc., are of high resolution and have centimetre-level accuracy as seen from Table 3 and Table 4, helping in the accurate estimation of volumetric changes of sediments. Moreover, the images obtained via aerial photogrammetry are orthorectified and mosaiced to form orthomosaics and used for delineating shoreline position to calculate shoreline changes [91,92,94,110]. UAVs are versatile and have been extensively used in the field of geoscience. The three review papers by [82,83,84] provide a comprehensive overview of UAVs for coastal monitoring.

As SfM is used extensively for the generation of DEM/point clouds/orthophotos/orthomosaics, it might benefit the user to get an idea of the commonly used software for this purpose, as shown in Figure 5. Agisoft Metashape is the most-used software for the generation of topographic data (four papers from terrestrial photogrammetry and seven papers from aerial photogrammetry). PIX4dMapper was found to be only exclusively used for processing UAV derived images (six papers). So, overall, Agisoft Metashape is the commonly used photogrammetric software for terrestrial photogrammetry, whereas for UAV derived images, both Agisoft Metashape and PIX4dMapper are used. Both of these software are proprietary. Another software called CloudCompare is frequently used to calculate morphological changes between point clouds. There are some open-source software such as MicMac and Visual SFM [88] for implementing SfM. However, a detailed intercomparison between these open-source software and the proprietary ones is beyond the scope of this present review.

The low-cost monitoring of topography through extracting variables such as DEM/DSM/point clouds is essential for the routine monitoring of nearshore topography, which is currently lacking, given that 24% of the world’s sandy beaches are eroding [6], limiting the predictive ability of numerical models for risk assessment of hazards [19].

Beach profiles using a conventional total station [143], an accelerometer-based wireless beach profiler that can also be mounted onto an autonomous robotic vehicle [145], or a system of GPS antennas/receivers mounted on a motor-quad help in the determination of shoreline changes and volumetric changes of sediments [144]. The latter two profiling methods are advantageous for low-cost monitoring compared to the former due to their mobility facilitating in the collection of data points, which also helps in the generation of a DEM. Sand level variation using geotagged photos of street signs that are used as ad-hoc erosion pins [73] and the wireless sensor network consisting of LDRs [116] are innovative interventions with the former, incurring no cost at all, but being dependent on the availability of pre-existing street signs while the latter is the only intervention within the reviewed papers showing promise in the real-time monitoring of volumetric changes with an accuracy of 5 cm, which is comparable to the DEM accuracies of terrestrial and aerial photogrammetric techniques. Erosion pins are found to be low-cost and effective for soil erosion monitoring [154,155] and thus could be easily set up to monitor erosion without the necessary presence of pre-existing street signs.

Wind is an essential parameter that needs to be monitored within coastal areas, as increasing onshore wind speeds during storms can cause considerable damage and also contribute to increased wave run-up and overtopping of coastal defences through processes such as wind-induced setup [156]. Using a low-cost wind instrument such as that of [150] would densify in situ onshore wind measurements near the coast and could be integrated with sand level monitoring systems like that of [116], as the authors acknowledge the possibility of integrating a wind instrument in the sensor node to better understand the correlation between wind parameters and sediment transport.

Shoreline position/shoreline proxies/waterline (categorised together as shown in Table 6) is a common variable derived from terrestrial and aerial photogrammetric products and is used to quantify shoreline changes/erosion as well as intertidal topography. Shorelines derived from VMS also form the basis for developing coastal state indicators (CSIs; [157]. Timex images from VMS [38,39,43,44,45,47,53,54] are used to detect shoreline position based on algorithms that can discriminate the intensity variability of the image at the land–water interface [32] such as semi-automatic and automatic image segmentation algorithms [53,54,158,159]. These VMS derived shorelines are not only used to measure shoreline changes but also the intertidal topography [38,53] based on the methodology developed by [160]. Ref. [75] demonstrated the effectiveness of the smartphone camera to accurately measure the shoreline following proper camera calibration. Surfcam-derived shorelines [69,70] using the transect method by the respective operating agency (see Section 3.1.3) returned inaccurate shorelines when compared against standard instruments, but demonstrated improvements upon the application of geometric correction techniques. It was also concluded by [69] that surfcams are better suited to measure shorelines than nearshore wave characteristics. Refs. [77,79] used time-lapse images from a time-lapse camera to monitor storm surges and changes in the intertidal morphology. While the former used waterline delineation to measure storm surge, the latter used the leading edge of an intertidal bar as a shoreline proxy to monitor intertidal geomorphological changes.

VMS, time-lapse cameras, and wireless VMS (also called video sensor networks), besides capturing the hazards, also capture the relevant forcing agents leading up to the hazards. However, Refs. [115,119], which deploy VSNs, do not provide a description of the methodology to derive the respective variable for coastal erosion monitoring.

Complementary approaches utilising the high temporal frequency of VMS and high spatial resolution of UAV photogrammetry to monitor shoreline changes and volumetric changes of sediments provide a synergistic monitoring of beaches, for instance, by accounting for the lower accuracy in the VMS derived intertidal beach profile by the high-resolution UAV derived DEM [110].

Such a complementary approach of integrating terrestrial and aerial photogrammetry is also utilised within the STIMARE project to monitor coastal morphological and shoreline changes as well as wave run-up [42].

Oftentimes, sea level, tides, waves, and storm surges are responsible for causing many of the observed coastal hazards. The observed sea level X(t) varying with time can be generally represented as [161]:(7)$$X\left(t\right)=Z_{0}\left(t\right)+T\left(t\right)+R\left(t\right)$$

Here, Z_0_(t) = mean sea level, T(t) = tidal part of the variation, R(t) = residual component.

Consistent monitoring of observed coastal sea levels and therefore the tides is essential to quantify the impact of sea level rise on coastal zones. Apart from in situ low-cost water level sensors that measure the absolute pressure [137,138], remote measurements can be carried out using IoT ultrasonic water level sensors [140] and by using GNSS-Reflectometry (GNSS-R; [111,112,113]) with low-cost GPS modules, which have shown promising results when validated against standard reference instruments (see Section 3.4 and Table 5). Such remote measurements are useful to ensure the longevity of the sensor, which otherwise is prone to corrosion, physical damage, and theft in situ [112]. GNSS-R is also useful, as it accounts for vertical land motion showing absolute trends in coastal sea levels [113], which is useful as many tide gauge records around the world are not tied to any GNSS [19]. For the in situ water level sensors that measure the water level (depth) based on the absolute pressure, the water level has to be barometrically corrected, implying the separate measurement of atmospheric pressure. Ref. [152] designed a prototype cable-mounted robot that eases the monitoring of data collection in difficult coastal environments during extreme weather events. This robot was a platform onto which a multiparameter sonde was attached to observe the variables of interest. Though [152] does not provide the data collected by this platform, the data are presented in a paper by [153] showing the ability of this system to measure multiple parameters, of which its ability to measure absolute and barometric pressure were of interest in this review, showing good agreement.

A number of low-cost sensors for measuring wave run-up and wave characteristics ensures the monitoring of wind waves, which are a major contributor to coastal hazards [162]. DIY Arduino-based wave gauges [134,135] have been shown to effectively monitor the nearshore waves when compared to standard gauges. The practical limit was on the sampling frequency, which was 4 Hz [135] and between 8 and 10 Hz [134] ,which excludes the detection of waves with a higher sampling frequency; however, this sampling rate can be adjusted. The in situ water level sensors [137,138] could be also used to measure nearshore waves [137] by changing the sampling interval. Low-cost GPS buoys [26,123,124,125] have been used extensively for measuring wave characteristics at high sampling rates. Validation with reference instruments has yielded an acceptable accuracy range [26,123,124] that helped establish the potential of these instruments for nearshore wave monitoring, although [123] states that off-the-shelf-GPS position receivers may not always yield the high data quality of the high-end Datawell wave rider buoys. VMS, too, has been utilised to study nearshore processes such as wave run-up [38,43,44,45] as described by [32]; however, no validation was provided. Finally, Ref. [151] used a UAV RTK-Lidar to measure waves and tides with acceptable accuracy, but this can be constrained according to the height at which the flight is flown.

With extreme weather events becoming more frequent, there is growing pressure to increase the resilience of coastal areas, which are hotspots of human settlement and economy, and given the dearth of sufficient monitoring equipment at those locations, the data scarcity created only adds to the growing difficulty of the responsible actor to make any useful decisions. It is urgent, therefore, that low-cost sensors that have been evaluated in the field and shown to perform within the acceptable standards be integrated as quickly as possible, especially in regions where the instalment of traditional instruments could be cost-prohibitive and time-consuming. The growing need for low-cost sensors is also reflected in the total increase of papers from 2016 to 2021 as compared to 2010–2015 within the search criteria, as shown in Figure 6 below.

As developing countries and small islands are at increasing risk from climate hazards, it would be beneficial for them to take advantage of the outputs of these existing low-cost interventions to generate a reliable and consistent output of marine data, which could help inform suitable and timely adaptation to protect their coasts. It has been found that the benefits of such proactive measures exceeds the social and economic costs of inaction [163].

## 5. Conclusions

It is unequivocal that the changing climate caused by increased greenhouse gases has negatively impacted the atmosphere and ocean, rendering densely populated coastal regions vulnerable to hazards such as coastal flooding and erosion due to climate-induced changes in various metocean variables such as sea level and waves. Due to the inertia in the ecosystem, these climate-induced changes are locked in for a certain duration, implying that even with drastic cuts in greenhouse gas emissions, human societies will continue to experience the negative impacts of the warming climate, making adaptation inevitable for developing climate resilience. As coastal monitoring equipment are sparse owing to the difficulty of setting up instrumentation and their high cost, relevant coastal data are sparse. This could limit timely implementation of relevant adaptation measures. Thus, this review was carried out to identify low-cost sensors from the peer-reviewed domain to monitor key coastal variables at a reasonable cost to fill the existing data gaps, and it was found that these coastal variables could be broadly mapped into three main coastal hazards: coastal flood, coastal erosion, and shoreline changes. This review, although not exhaustive, successfully demonstrated that different low-cost sensors exist for monitoring key coastal variables and most of these sensors have been validated against a high-cost reference instrument showing promising results in the field environment. Several low-cost sensors exist to measure similar variables (for instance, to measure topography), facilitating the end user to make monitoring choices according to their specific needs. The accuracy reported is a guide and not aimed at facilitating the end user to make a final decisive choice, but rather to make the user cognizant of the available low-cost instruments within the peer-reviewed domain that can be explored and adapted to meet their specific needs so that the cost of the traditional monitoring sensors does not limit timely implementation of adaptation interventions in the most vulnerable coastal regions of the world. Though significant progress has been made in Earth observation of coastal hazards and the corresponding forcing agents, the same cannot be said for large-scale in situ low-cost monitoring that facilitates data collection at a fine spatial and temporal scale, which is a limitation of Earth observation products. The secondary research conducted in this paper reveals that low-cost sensing techniques based on existing physical principles can be exploited to collect in situ data within acceptable standards (in comparison with reference high-cost instruments), creating the opportunity to build a dense network of such low-cost sensors that can help in the calibration and validation of various forecasting tools such as storm surge forecasting, as well as help in the validation of coarse EO products. Some low-cost products such as the WSN for storm surge measurements [117] could help in the near-real-time measurements of variables such as water level, which could be potentially integrated into an “early warning support” for coastal communities, for instance. Therefore, real-time monitoring of parameters such as water level, tides, waves from various low-cost sensing techniques such as GNSS-R, using Arduino-based pressure gauges, low-cost industry standard water level sensors, etc., could be considered. While variables such as nearshore topography can be monitored in real time using VMS, the product of interest, however, from such monitoring, which is the DEM/DSM, can only be derived after post-processing of the acquired images. So, even though to the best of our knowledge at the time of writing this paper, such topographic data cannot be derived in “real time”, low-cost interventions using UAVs and VMS can help monitor it at a sufficiently high temporal scale. UAVs have been shown to derive topographic data with an extremely high accuracy at the centimetre scale. Such high-resolution spatial data facilitates the accuracy of numerical modelling efforts. Thus, through this systematic review, the identification of various low-cost sensors provides an opportunity to deploy and evaluate such interventions for the large-scale in situ monitoring of coastal hazards, their drivers, and the coastal zone characteristics at a fine spatial and temporal resolution and could be useful interventions in areas currently lacking monitoring equipment.

## Figures and Tables

**Figure 1 sensors-23-01717-f001:**
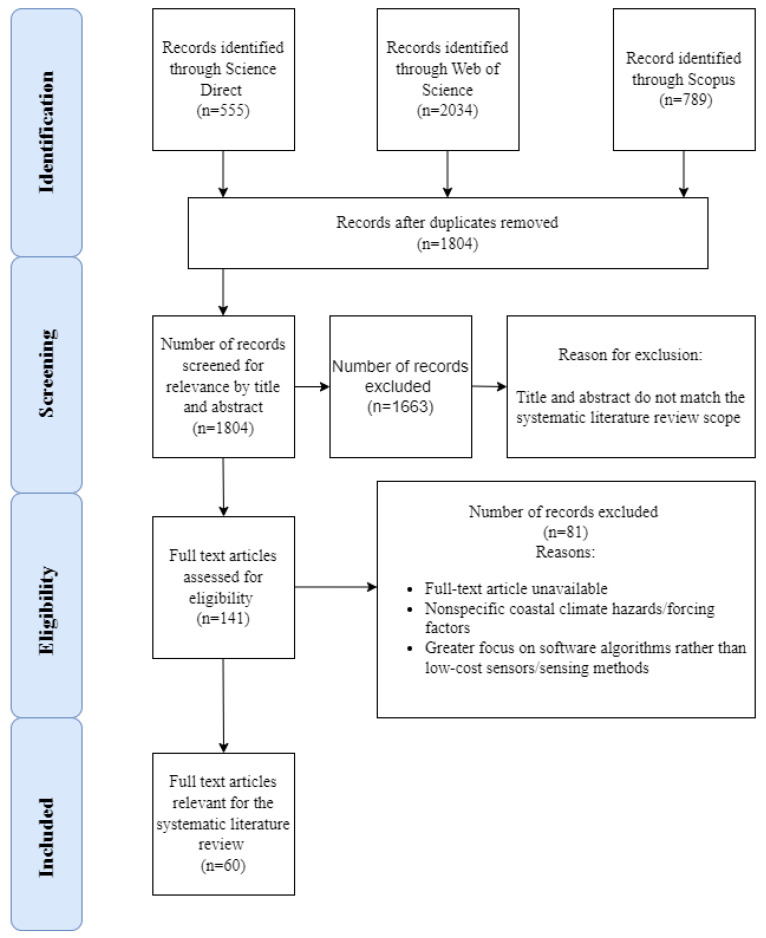
PRISMA diagram for the systematic literature review on low-cost sensors for monitoring coastal climate hazards.

**Figure 2 sensors-23-01717-f002:**
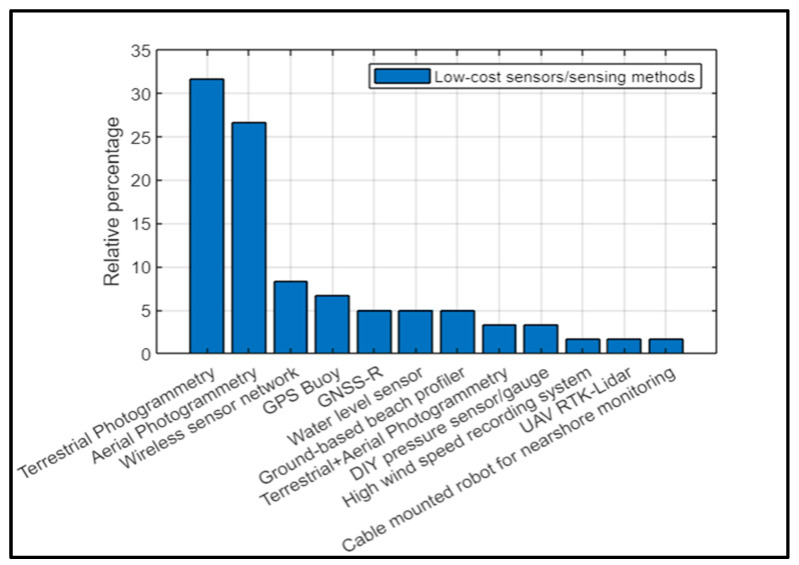
Percentage of the low-cost sensors/sensing methods.

**Figure 3 sensors-23-01717-f003:**
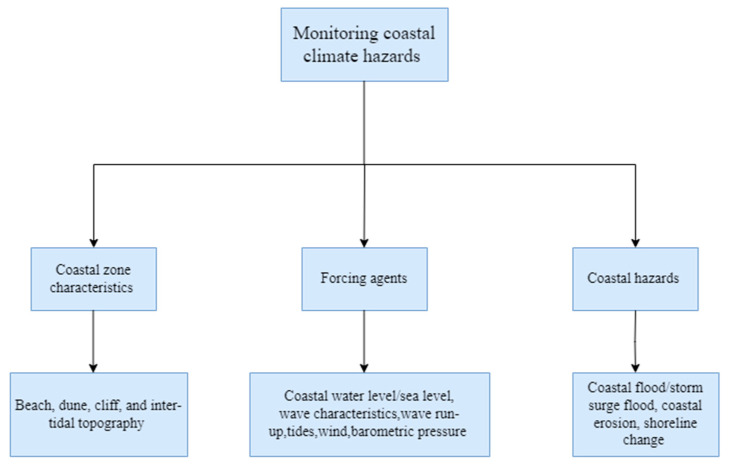
The coastal zone characteristics, metocean variables, and coastal hazards monitored by the sensors reviewed in this paper.

**Figure 4 sensors-23-01717-f004:**
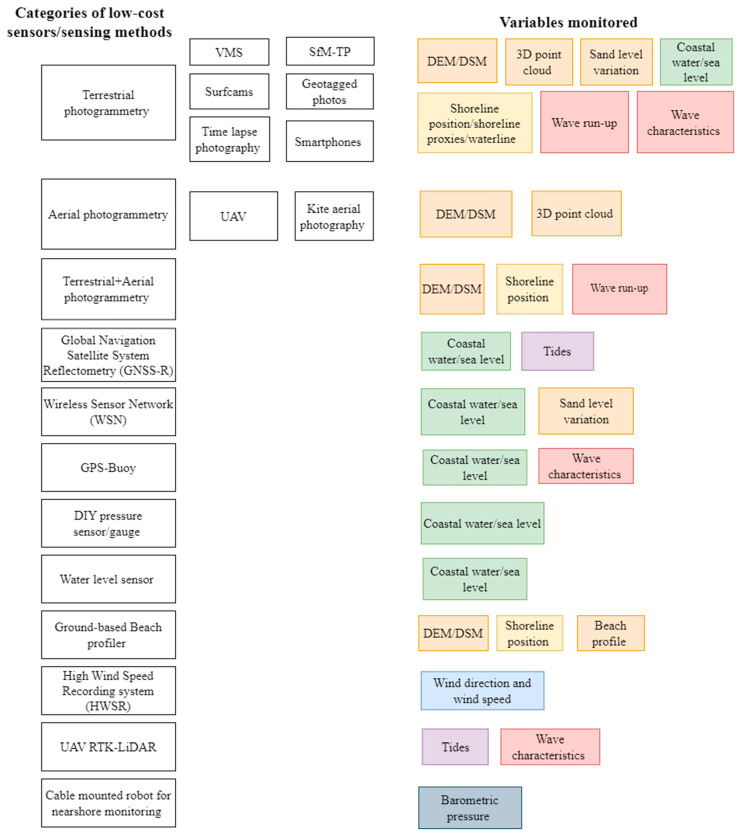
The different variables obtained from the sensors/sensing methods; variables have been colour coded, with similar variables being coloured the same.

**Figure 5 sensors-23-01717-f005:**
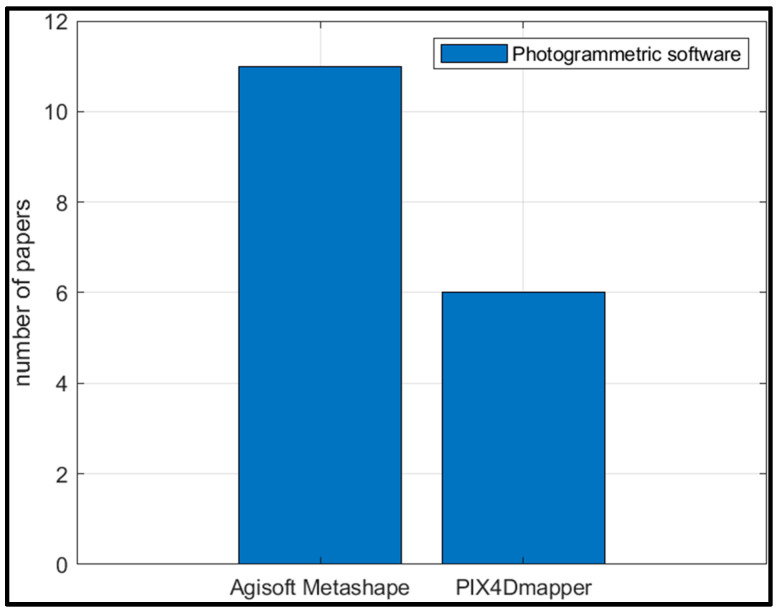
Photogrammetric software employed for the extraction of DEM/DSM/point clouds/orthophoto/orthomosaics.

**Figure 6 sensors-23-01717-f006:**
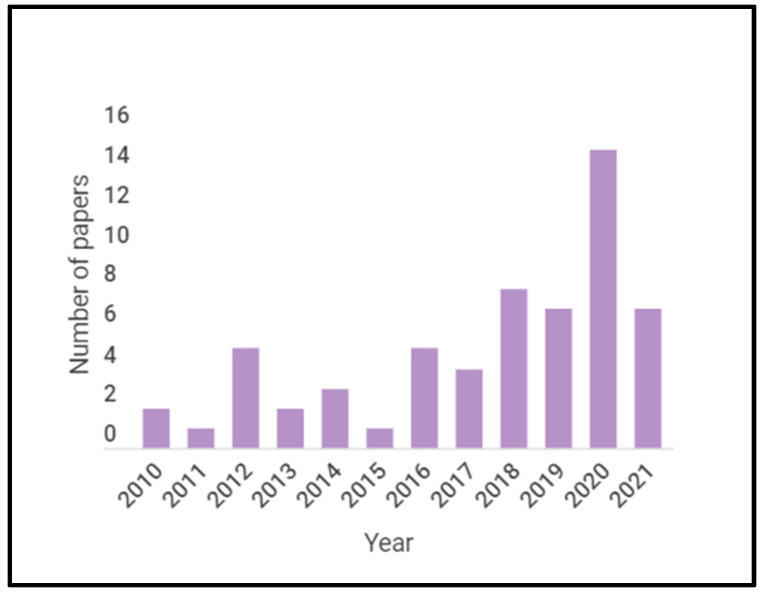
Distribution of the 60 papers during 2010–2021.

**Table 1 sensors-23-01717-t001:** Combination of keywords for retrieving relevant articles from three scientific databases (Web of Science, ScienceDirect, Scopus).

Keyword Search Strings	Science Direct	Web of Science	Scopus
(coast OR coastal OR hazard OR climate) AND (low-cost sensors OR citizen science sensors) AND NOT Air quality	140	102	39
Monitoring AND low-cost AND (coast OR coastal OR hazard)	195	622	719
(low-cost sensors OR citizen science sensors) AND (coastal OR coast OR climate OR erosion OR flooding OR storm surge OR sea level rise)	124	776	1
(climate OR coast OR coastal) AND (low-cost sensors OR citizen science sensors) AND NOT air-quality	96	534	30
Total	3378		

**Table 2 sensors-23-01717-t002:** Inclusion/exclusion criteria for the selection of relevant articles from the systematic search of peer-reviewed literature.

Criterion	Inclusion	Exclusion
Coastal hazards	Coastal flooding Coastal erosion Beach, dune, and cliff erosion Storm surge Shoreline changes	Coastal hazards not included in the inclusion criteria, such as maritime security hazards, marine pollution, marine ecosystem shifts, coastal landslides/slope stability, tsunamis, compound floods, flash floods
Forcing agents	Sea level/water level Surface wind Surface wind-wave Tide Extreme events such as stormsBarometric pressure (related to storms)	Forcing agents not included in the inclusion criteria such as vertical land motion, land cover and land use, fluvial sediment supply, river discharge, ground water level, sea surface temperature, precipitation
Coastal zone characteristics	Coastal topography Intertidal topography	Bathymetry
Type of sensors	Low-cost sensors or citizen science sensors	High-end sensors, unsuitable for citizen science activities
Nature of Hazard	Climate-induced	Not related to climate or not climate-induced
Geographical coverage	Coastal regions other than the Arctic (except for coastal regions inhabited by humans) and Antarctic, as these are exceptionally harsh environments requiring special types of sensors	Arctic and Antarctic

## Data Availability

Not applicable.

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
