# Peer review of "Low-Cost Sensors for Monitoring Coastal Climate Hazards: A Systematic Review and Meta-Analysis"

_sensors, 2023, doi:10.3390/s23031717_

Round 1

Reviewer 1 Report

In the manuscript under consideration, the authors conducted a review of low-cost techniques (based on sensors) developed and used in the last decade for monitoring coastal hazards and their forcing agents. Despite the fact that the manuscript presents a topic of potential interest to the readers of the journal, there are several issues to be addressed before the manuscript is considered further. Here are some of the issues:

·       All the equations should be properly formatted, labeled, and referenced in the text.

·       The study is based on two major issues: Low-cost sensors and coastal climate hazards. However, the introduction has mainly focused on the coastal climate hazards with little covered about the low-cost sensors. Hence, it is suggested that the authors provide a sufficient base of low-cost sensors in the introduction section. It is important that the authors clearly define what is a low-cost sensor in this context. The following should be covered:

ü  Some typical examples of coastal climate hazards reported in the literature.

ü  The meaning of low-cost sensors.

ü  General types of low-cost sensors.

ü  Advantages and disadvantages compared to other sensors.

·       The authors should consider improving the quality of the text in the figures.

·       The bolding of figures and tables references in the text should be removed.

·       The referencing in the manuscript should be done in accordance with the guidelines.

·       Generally, the biggest flaw I see in this work is the fact that the “low-cost” part has not been well explained in the manuscript. What makes the highlighted sensors low-cost or the “low-cost” is based on what factors? Operation cost (e.g., workforce and energy consumption)? Maintenance cost? Purchasing price? Durability (no need for frequent replacements)? There should be a very clear description of that. It should also be noted that something of “low-cost” can be regionally dependent. May be termed as “low-cost” in developed countries but may not be the case in developing countries.

·       All the abbreviations in the manuscript should be defined before their application elsewhere.

Author Response

Dear Reviewer,

Thanks very much for your invaluable comments.

Reviewer 2 Report

The manuscript (sensors-2167913):”Low-cost sensors for monitoring coastal climate hazards: a systematic review and meta-analysis” gives an overview of broad range sensor technologies used for monitoring the relationship between coastal behavior and climate change. The sections are systematically arranged and different low-cost methodologies provided from a decent number of relevant papers are well elaborated.

The manuscript is well written and structured. However, there is considerable amount of excessive explanations and it can be improved.

Major concerns.

In general, the main purpose of review articles is to summarize the most relevant papers focused on specific research field and based on inter-study comparison to draw the conclusions which can guide other researchers. Also, the aim is not to give too many details like in monographs. Accordingly, technical aspects of selected sensor methodologies in this manuscript, such as procedure description and software solutions should be brief and concise. Otherwise, the reader’s attention will be out of focus.

Finally, I recommend the authors to remove at least 10% of text from the section 3. to be more clear and easier to read.

Author Response

(The authors gave the same response as above.)

Reviewer 3 Report

I read with great interest the manuscript submitted by Ahmed et al. for consideration of Sensors. This manuscript presents a great review of low-cost sensors for monitoring coastal climate hazards: a systematic review and meta-analysis. Although the manuscript presents a good dataset and addresses relevant research questions, I consider that it cannot be accepted for publication in its present form. Please see my detailed comments below.

General

I think the citation style is not in accordance yet with the format of the journal.

Introduction

-          Lines 74-75: Please add references to land subsidence as other factors that may worsen the coastal flood.

-          Line 104: Please elaborate on what the authors mean by "the latest".

-          L105: Please elaborate on "various hazards". Is it all coastal hazards or only specific hazards?

Methods

-          L111: Please elaborate on “27 items”.

-          L114: In tourism à How about in disaster studies? It is more relevant to the topic.

-          L116: Why did the filter only specific for those databases (Scopus, SD, and WoS) and did not consider another database such as ACM and IEEE? Are there any specific reasons?

-          L117: Why from 2010? Please elaborate.

-          L139-140: I think it will be interesting if the authors can mention how many articles they excluded due to the "barrier" existing to full-article and how they manage the critical information that may exist in those articles.

-          Fig. 1: I think an arrow is missing between step 1 and 2.

Results

Please summarize your results to be more concise, compare each other, and only show the most exciting and essential findings rather than explain their research one by one in an enormous long text (37 pages!).

Discussion

L1623-1636: This statement is too early and may be biased since the reason is maybe due to several restrictions used by the authors since the beginning, i.e., English language and accessible articles.

----------------------------

Author Response

(The authors gave the same response as above.)
